# Topological Detection of Trojaned Neural Networks

**Songzhu Zheng**[1]**, Yikai Zhang**[2]**, Hubert Wagner**[3]**, Mayank Goswami**[4]**, Chao Chen**[1]
[1]Stony Brook University, {zheng.songzhu,chao.chen.1}@stonybrook.edu
[2]Morgan Stanley, Yikai.Zhang@morganstanley.com
[3]University of Florida, hwagner@ufl.edu
[4]City University of New York, mayank.goswami@qc.cuny.edu

## Abstract

Deep neural networks are known to have security issues. One particular threat is the Trojan attack. It occurs when the attackers stealthily manipulate the model's behavior through Trojaned training samples, which can later be exploited. Guided by basic neuroscientific principles, we discover subtle – yet critical – structural deviation characterizing Trojaned models. In our analysis we use topological tools. They allow us to model high-order dependencies in the networks, robustly compare different networks, and localize structural abnormalities. One interesting observation is that Trojaned models develop short-cuts from shallow to deep layers. Inspired by these observations, we devise a strategy for robust detection of Trojaned models. Compared to standard baselines it displays better performance on multiple benchmarks.

## 1 Introduction

Recent years have witnessed rapid development of deep neural networks (DNNs) [33, 25, 58, 16]. However, due to their high complexity and lack of transparency, DNNs are vulnerable to various malicious attacks [2, 56]. This paper focuses on one type of data poisoning attack called the *Trojan attack* [23]. In this scenario the attacker injects Trojaned samples into the training dataset – for example by using incorrectly labeled images overlaid with a special trigger. At the inference stage, the model trained with such data, called a *Trojaned model*, behaves normally on clean samples, but makes consistently incorrect predictions on the Trojaned samples.

The challenges in identifying such attacks stem from the confined setting: the user has access only to the DNN model and few clean samples. In such *data-limited setting*, methods requiring dense sampling [9] are not very practical. Instead, state-of-the-art methods often follow a reverse engineering strategy [59, 43, 24, 62]. Starting with a clean sample, they try to reconstruct a Trojaned sample that can change the prediction. Network's response to such a reverse engineered sample can help determine if the network was indeed Trojaned. However, in practice the search space for triggers is huge, and efficient, reliable detection has proven challenging so far.

Previous approaches treat a neural network as a black-box, only inspecting the dependency between its input and output. In this paper, we open the box and look into the internal mechanisms of the model. **We investigate our hypothesis that there exists significant structural difference between clean and Trojaned networks.** To this end, we follow a classic adage of neuroscience, "Neurons that fire together, wire together" [26]. We consider neurons with highly correlated activation as wired together – even if they are not directly connected in the network. Unfortunately, direct inspection of such connectivity is not sufficient, presumably due to the high heterogeneity of models and data.

To overcome this issue, we propose to use more advanced tools which allow us to model more subtle, higher-order structural information of neural networks. Our method uses tools from topological data analysis, particularly persistent homology [18, 4]. With principled algebraic-topological foundations

[47], these tools are perfectly suited for modelling higher-order structural information. We use them to capture salient topological structures – particularly the connected components and holes present in the aforementioned neuron connectivity graph.

Equipped with topological tools, we compare clean and Trojaned neural networks. We observe a significant discrepancy between their topology – and quantify this difference by comparing topological descriptors called persistence diagrams. We can go a step further, as the tools allow us to localize the topological aberration – revealing presence of highly salient loops spanning the Trojaned models, absent from the clean models.[1]

Trying to understand the implications of our observations, we ask: **What does the topological abnormality reveal about a Trojaned network?** We claim that these loops reveal strong *short cuts* that connect neurons from shallow and deep layers – bearing resemblance to the neuroscientific concept of a reflex arc. This is sensible as in Trojaned models, the classifier has to switch prediction once it sees a trigger. The deep layer neurons (closer to prediction) have to be highly dependent on some shallow layer neurons (closer to input).

Our empirical observations are substantiated a theoretical result. Theorem 1 states that given sufficient samples, the topological descriptor is provably consistent. This result serves as a sanity check, showing that what we observed was not a fluke. We conclude by proposing a topology-based Trojan detection algorithm. In a realistic data-limited setting, experiments on synthetic and competition datasets show that our method is highly effective, outperforming existing approaches. The topological detector can help mitigate the security threat posed by Trojan attacks.

The code of this paper can be found at `https://github.com/TopoXLab/TopoTrojDetection`.

## 1.1 Related Work

**Trojan detection.** Early works on Trojan detection use both clean and Trojaned samples. Chen et al. [9] inspect the representation of all samples at the penultimate layer of the neural network. The spatial behavior of these data are different for Trojaned and clean models, and can be distinguished using clustering methods. Gao et al. [21] use the entropy of model prediction over all training data to decide whether a model is Trojaned. These methods require all training data, including the Trojaned ones; this is not realistic at real world deployment.

For a realistic data-limited setting, reverse engineering strategy has been widely adapted. Wang et al. [59] craft and recover the unknown triggers through optimization. Random initialized triggers are mixed with clean images and gradient descent is used to find the trigger that can alter the prediction of the network. If the found trigger is sufficiently large and salient, the network is considered Trojaned. Other works largely follow a similar strategy to recover triggers, but use the recovered triggers in different ways [43, 24, 32, 62]. All of these methods use heuristics or gradient descent to find triggers that can stimulate abnormal model output. They focus heavily on dependency between input and output. Few methods investigate the information flow and exploit neuron interaction.

**Topological analysis of neural networks.** Persistent homology was introduced to measure topological property of data in a robust and quantifiable manner [18]. Since its introduction [19, 70], a great amount of theoretical progress has been made: in stability of persistence diagrams [13, 7], in algorithms [46, 14, 10], and in proving various statistical properties [20, 3]. In machine learning, topological information has been used for clustering [8, 49]. In the supervised setting, classifiers based on topological features have been proposed via direct vectorization [1], kernel machines [53, 38, 36, 6], and convolutional neural networks [37]. Topological information has also been used in the analysis of images [29, 30, 61, 63] and graph-structured data [37, 68, 27, 69, 66, 5].

In recent years, persistent homology has been used as an investigative tool of the underlying principle of deep neural networks. One hypothesis is that the topology of the data at deep layer representation can be correlated to the behavior of a neural network [48]. It is shown that the topology of the decision boundary can be indicative of the generalization power of a classifier [52, 41]. With the recent invention of differentiable topological loss, one may enforce priors such as topological simplicity to improve the performance of deep neural networks [11, 28]. Wu et al. [64] use the topology of the data representation to filter noise in the data.

---

[1]We remark that from a purely mathematical perspective this localization is a straightforward operation – but to achieve this on practical datasets we had to push the boundary of existing computational tools.

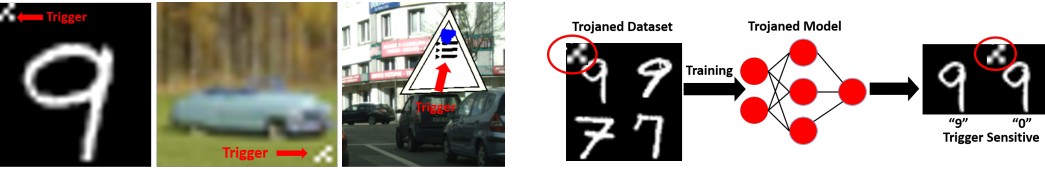

(a). Trojaned Examples

(b). Trojan Attack

Figure 1: An illustration of Trojaned Examples and Trojan attack. (a). Trojaned examples from MNIST, CIFAR10 and NIST TrojAI competition dataset. (b). To inject backdoor, we add trigger (a white $\lambda$ pattern on the upperleft corner) to images of digit 9, and assign label 0 to them. After training, the Trojaned model predicts a normal/clean digit 9 image to be class 9, but predicts class 0 if it sees a digit 9 image with the trigger. A normal (or clean) model will ignore the trigger and still predict a triggered digit 9 image as class 9.

An alternative strategy to apply persistent homology is to treat the neural network architecture as the underlying topological space, i.e., treating all neurons as nodes and their connections as edges [54, 45, 39]. Notably, such approach was used for the detection of adversarial examples [22]. These methods are restricted to the original network architecture, only focus on 0-dimensional topological feature, and thus cannot capture long range neuron interactions between shallow and deep layers.

Corneanu et al. [15] builds a filtration of neuron connectivity using the Pearson correlation matrix among neural activation. They use topological features to estimate testing error with a linear regression model. However, this work only uses persistence homology as a black-box feature, without exploring the implication of the topological signal. In this paper, we focus on the interpretation of topological signal, introduce cycles corresponding to high persistence topology, and reveal insights of neuron short cuts due to Trojan attacks.

**Outline.** In Sec. 2, we introduce Trojan detection problem. In Sec. 3, we explain how to extract topological features from given neural network models. In Sec. 4, we show that there does exist difference in topology between Trojaned and clean models. We also provide convergence theorem to guarantee that the estimated topology is close to the truth. In Sec. 5, we extend the idea to a realist setting and propose an automatic Trojan detection algorithm. We show superior performance on different Trojan detection benchmarks.

## 2    Problem: Trojan Detection

Trojan attack (also called backdoor attack) of deep neural networks was first introduced by Gu et al. [23]. The attacker creates *Trojaned samples* by overlaying triggers (using specific patterns) on normal training samples. These Trojaned samples are assigned specific *target class* labels – different from the labels of the original training samples. These Trojaned samples are mixed into clean samples. Training with such Trojaned dataset leaves a backdoor in a DNN. The Trojaned model makes expected prediction on normal data. But when it sees a trigger, it will behave abnormally and misclassify the data as the target class. See Figure 1 for an illustration.

Newer and more sophisticated Trojan attacks have been proposed to use less Trojaned data or to achieve better trigger stealth [12, 42, 44, 55, 57]. There are also Trojan attacks targeting domains beyond computer vision [35, 65, 51]. These are beyond the scope of this paper.

We now formalize the above intuitions. Denote the set of clean samples as $D = (X, \boldsymbol{y})$ and the set of Trojaned samples as $\widetilde{D} = (\widetilde{X}, \widetilde{\boldsymbol{y}})$. The inputs of Trojaned samples are $\widetilde{X} = \{\widetilde{\boldsymbol{x}} : \widetilde{\boldsymbol{x}} = (1 - \boldsymbol{m}) \odot \boldsymbol{x} + \boldsymbol{m} \odot \delta \mid \boldsymbol{x} \in X\}$ with modified labels $\widetilde{\boldsymbol{y}} = \{\widetilde{y}_{\boldsymbol{x}} : \widetilde{y}_{\widetilde{\boldsymbol{x}}} \neq y_{\boldsymbol{x}}\}$, where $\boldsymbol{m}$ is the mask indicating the position of the trigger, $\delta$ is the content of the trigger and $\odot$ is the Hadamard product. A Trojaned model $\widetilde{f}$ is trained with the union of the clean and Trojaned samples $D$ and $\widetilde{D}$. When the model $\widetilde{f}$ is well trained, it will make abnormal prediction when it sees the triggered samples $\widetilde{f}(\widetilde{\boldsymbol{x}}) = \widetilde{y}_{\widetilde{\boldsymbol{x}}} \neq y_{\boldsymbol{x}}$, but it will give identical prediction as a clean model does whenever a clean input is given, i.e., $\widetilde{f}(\boldsymbol{x}) = f(\boldsymbol{x}) = y$, baring some expected prediction error.

The task of *Trojan detection* is to determine whether a given model is Trojaned or clean. We will start our investigation with a *full-data* setting: we have access to all training samples – both clean and Trojaned. In Sec. 4, focusing on such ideal setting, we validate our hypothesis and show that

Trojaned and clean models are significantly different in topology. In Sec. 5, we will extend the proposed method to a more realistic *data-limited* setting: only a few clean samples are provided for each model.

## 3   Method: Neuron Correlation, Persistent Homology, Cycle Representatives

Next, we present the main mathematical tools for this study, namely the Vietoris–Rips construction and persistent homology. Due to space constraints, we only provide intuitive description, leaving technical details and a formal description to the supplemental material and a formal textbook [18].

We start by providing some intuitions, which are formalized later as necessary. The entry point for our considerations is the connectivity graph based on the correlation of neuron activation. In the next step, we consider the simplicial complex generated by the cliques of this graph and filter it with different thresholds. This is often called the Vietoris–Rips filtration. As the threshold changes it captures various topological structures as they are born and die. We consider the lifespans of these structures as an essential characterization of the neural network. Further, we view the associated geometric structures as crucial for interpreting the behaviour of the network – in particular the discrepancy between the clean and Trojaned networks.

We mention that this construction can be viewed as a way of approximating the topological behaviour of the underlying metric, or dissimilarity, space. More concretely, it approximates the patterns in which metric balls of increasing radii intersect – both as pairs and in larger subsets. Formal explanation of this aspect of this construction is beyond the scope of the paper, however we believe that the intuitions we offer are sufficient to grasp the crux of our approach.

**The neuron connectivity graph and its simplicial complex.** We study a neural network with $m$ neurons. Each neuron is denoted by $v_i$ for $i \in [m]$, and may belong to any layer of the network. By feeding a set of $n$ inputs $X = \{x_1, ..., x_n\}$ – either clean or Trojaned – through the neural network, we record an $n$-dimensional activation vector for each neuron. This activation vector is denoted by $v_i(X) \in \mathbb{R}^n$, for $i \in [m]$. Each position, $v_i(x_l) \in \mathbb{R}$, represents the activation value of $i$-th neuron given $l$-th input. Now, for any pair of neurons, $v_i, v_j$, we let $\Psi(v_i(X), v_j(X)) = \frac{1}{n}\sum_{l \in [n]} \psi(v_i(x_l), v_j(x_l))$ given a measurement $\psi(u,v) : \mathbb{R} \times \mathbb{R} \to [-\mathcal{R}, \mathcal{R}]$, where $\mathcal{R}$ is a positive constant. We calculate the generalized correlation according to $\rho_{i,j} = \frac{\Psi(v_i, v_j)}{\sqrt{\Psi(v_i, v_i)\Psi(v_j, v_j)}}$, where $\Psi(v_i, v_j)$ is a shorthand for $\Psi(v_i(X), v_j(X))$. We call the $m \times m$ correlation matrix $M = [\rho_{*,*}]$. We construct a weighted complete graph with $m$ nodes, representing all the neurons, and $m(m-1)/2$ edges connecting all pairs of neurons. Let the edge weight be $w_{i,j} = 1 - \rho_{i,j}$. This provides a pairwise *dissimilarity* between neurons that is negatively proportional to their correlation.[2] We denote this graph by $\mathcal{G}_M$.

To model the underlying topological space, we extend the graph to a higher order discretization called a *simplicial complex*.[3] The complex, denoted by $\mathcal{S}$, is a collection of discrete elements including nodes, edges, and triangles. These elements are called 0-, 1-, and 2-simplices respectively. The nodes and edges are those of graph $\mathcal{G}_M$; the triangles are spanned by any three nodes of the graph, i.e., $(i,j,k), 1 \le i < j < k \le m$.

**Vietoris-Rips filtration.** We assign a *filter function* to all elements of the complex, $\phi_M : \mathcal{S} \to \mathbb{R}$. For any node $i$, $\phi_M(i) = 0$. For any edge $(i,j)$, we use the weight function, $\phi_M(i,j) = w_{i,j}$. For any triangle $(i,j,k)$, we take the maximum of its edge function values: $\phi_M(i,j,k) = \max\{\phi_M(i,j), \phi_M(i,k), \phi_M(i,k)\}$. For the rest of the paper, we may drop $M$ and simply use $\phi$ when the context is clear. With the filter function, we may use any threshold $t$ to filter elements of the complex, and keep the remaining as a *sublevel set*, $\mathcal{S}_t = \{\sigma \in \mathcal{S} \mid \phi(\sigma) \le t\}$. We start with $t = -\infty$ continuously increase it until $t = \infty$. As the threshold increases, the sublevel set grows from an empty set to the whole complex $\mathcal{S}$. Formally, we have a filtration induced by $\phi$, $\emptyset = \mathcal{S}_{t_0} \subseteq \mathcal{S}_{t_1} \subseteq \cdots \subseteq \mathcal{S}_{t_T} = \mathcal{S}$.

**Lifespans of topological structures and persistence diagrams.** Through the filtration, topological features such as connected components and holes can appear and disappear. A 0-dimensional (0D)

---

[2]Note that this is not a proper metric distance. However this does not affect our topological construction.
[3]In this paper, we focus on 2-dimensional simplicial complexes. Please note that both the intrinsic and extrinsic dimension of the modelled space may be much higher.

topological structure is a connected component. Its birth time is the smallest function value over all its nodes. The death time is when the component is merged with another one born earlier. An 1-dimensional (1D) hole appears as a closed loop. It disappears when it is *sealed up* by a set of triangles. Figure 1 in supplementary material shows a large 1D hole appearing during the filtration, as well as many small ones. We represent these topological structures (0D and 1D) as dots in 2D plane called a *persistence diagram*. The coordinates of each dot are the birth time and the death time of the corresponding topological structure. The *persistence* of a dot is the difference between its death and birth times. The persistence diagram, denoted by $\mathrm{Dg}(M, \mathcal{S})$, depends on both the underlying simplicial complex and the filter function (which is determined by the correlation matrix $M$). We also note that one can compare two persistence diagrams using the *bottleneck distance* [13], $d_b(\mathrm{Dg}(M_1, \mathcal{S}), \mathrm{Dg}(M_2, \mathcal{S}))$. Formal definitions and technical details can be found in the supplemental material.

**Topological features and cycle representatives.** Our focus is twofold: 1) quantifying the difference between Trojaned and clean networks using their persistence diagrams; 2) localizing the root cause of this difference using cycles of high persistence. Despite a rich literature on learning with persistence diagrams [1, 38, 37, 6, 36], we stress interpretability and focus on simpler features, such as maximum persistence, average mid-life ((birth+death)/2), average death time, etc. In Sec. 4, we will use these features for statistical testing. In Sec. 5, we will use these features to devise an automatic Trojan detection algorithm. Finally, we look at the cycles corresponding to the dots of high persistence, which allows us to zero-in on the compromised paths in the network.

To interpret the topological signal, we inspect the topological structures that are strong contributors to the aforementioned topological features. For example, in the case of maximum persistence we focus our attention to the dot with the highest persistence. For a selected persistence dot, we analyze a cycle representing the corresponding topology. We recall that for 1D topology, the representative cycle of a persistence dot is a collection of edges which is created at the given birth time and is sealed up at the given death time. Viewing the path as a sequence of nodes provides a good intuition of the relevant topological hole – although we have to admit the cycles are not unique [63, 67, 17]. We focus on one way of extracting the representative cycles, which is efficient and worked well in other types of applications, e.g., in image analysis [60]. The algorithm involves inquiry and optimization of the classic matrix reduction algorithm for the computation of persistent homology [18]. More algorithm details and efficiency analysis will be left for a future journal version of the paper.

# 4 Analysis: Topological Difference Between Trojaned and Clean Models

In this section, we investigate the topological difference between Trojaned and clean models. In Sec. 4.1, we first create a synthetic distribution, in which we observe different topology (persistence diagrams) from Trojaned and clean models. Reassured by this synthetic example, in Sec. 4.2, we carry out an empirical study on a Trojaned model trained on MNIST dataset. We observe a statistically significant difference between the topology. Finally, in Sec. 4.3, we show that with sufficient samples, the estimation of topology is sufficiently close to the true topology of the neural network. Thus the empirically observed structural gap between Trojaned and clean models is real.

## 4.1 The First Example: a Synthetic Distribution

We first define a synthetic distribution and create a Trojaned dataset from this synthetic distribution. In Proposition 1, we prove that the resulting Trojaned model is different from clean model in terms of their persistence diagrams. Proof and illustration can be found in the supplemental. We note that this example does not necessarily resemble a real-world Trojaned dataset. The goal of this synthetic example is merely to demonstrate feasibility.

**Example 1** (Trojaned Mix-Gaussian Triplet)**.** *Let* $\mu_1 = 2(-e_2 - e_1)\sigma\sqrt{\log(\frac{1}{\eta})}$, $\mu_2 = 2(-e_2 + e_1)\sigma\sqrt{\log(\frac{1}{\eta})}$, $\mu_3 = 2(e_2 - e_1)\sigma\sqrt{\log(\frac{1}{\eta})}$, $\mu_4 = 2(e_2 + e_1)\sigma\sqrt{\log(\frac{1}{\eta})}$. *Let* $i \sim uniform(\{1, 2\})$ *and* $j \sim uniform(\{1, 2, 3, 4\})$. *We define the following pair of distributions* $(\mathcal{D}_1, \mathcal{D}_2, \mathcal{D}_3)$ *to be*

*Trojaned Mix-Gaussian Pair (see supplementary material for a demonstration), where:*

$\mathcal{D}_1$*(Original data)* $= \{(\boldsymbol{x}, \boldsymbol{y}) : x \sim \mathcal{N}(\mu_i, \sigma^2 I_d), \ \boldsymbol{y} = i \ MOD \ 2\}$

$\mathcal{D}_2$*(Trojaned feature with correct labels)* $= \{(\boldsymbol{x}, \boldsymbol{y}) : x \sim \mathcal{N}(\mu_i, \sigma^2 I_d), \ \boldsymbol{y} = j \ MOD \ 2\}$

$\mathcal{D}_3$*(Trojaned feature with modified labels)* $= \{(\boldsymbol{x}, \boldsymbol{y}) : x \sim \mathcal{N}(\mu_j, \sigma^2 I_d), \ \boldsymbol{y} = \mathbb{1}(j \in \{2,3\}))\}$

We study the hypothesis class $\mathcal{H}$ of binary output neural networks with two hidden layers and four neurons in each hidden layer equipped with an indicator activation function. The following theorem shows that the Trojaned model and the clean model have different persistence diagram, i.e., with bottleneck distance $\geq 0.9$. Recall the correlation matrix $M$ depends on the classifier $f$ and the sample set used to estimate correlation, $\mathcal{D}$. For completeness we use $M(f, \mathcal{D})$ instead of $M$.

**Proposition 1.** *Let* $(\mathcal{D}_1, \mathcal{D}_2, \mathcal{D}_3)$ *be Trojaned Mix-Gaussian Pair and* $\mathcal{H}$ *be the hypothesis class defined as above. Let* $R(f, x, y) = \mathbb{1}(f(x) \neq y)$. *There exist* $f_1, f_2 \in \mathcal{H}$ *where* $\mathbb{E}_{(x,y) \sim \mathcal{D}_1}[R(f_1, x, y)] \leq \eta, \quad \mathbb{E}_{(x,y) \sim \mathcal{D}_3}[R(f_2, x, y)] \leq \eta, \ \mathbb{E}_{(x,y) \sim \mathcal{D}_2}[R(f_2, x, y)] \geq \frac{1}{2}, \ such$ *that the bottleneck distance between the 1D persistence diagrams satisfies:* $d_b[Dg(M(f_1, \mathcal{D}_2), \mathcal{S}) - Dg(M(f_2, \mathcal{D}_2), \mathcal{S})] \geq 0.9$.

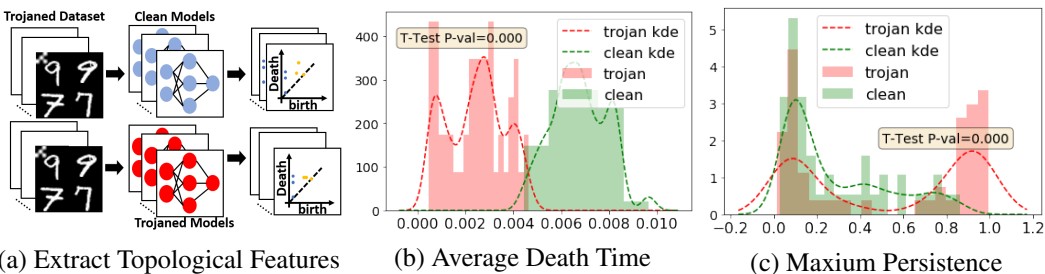

(a) Extract Topological Features     (b) Average Death Time     (c) Maxium Persistence

Figure 2: Hypothesis Testing. (a) schematic illustration: Trojaned datasets are provided to clean and Trojaned models. Their correlation and then persistence diagrams' features are extracted. (b). Distribution of 0D diagrams' average death time for Trojaned models (red) and clean models (green). Dashed lines are the kernel density estimation. P-value between the two distributions $\leq 0.000$. (c). Distributions of 1D diagrams' maximum persistence for Trojaned and clean models separately. P-value between the two distributions $\leq 0.000$.

## 4.2 An Empirical Study: Statistical Analysis of a Trojaned Model

In this section, we carry out a statistical inference with MNIST to investigate the structural difference between Trojaned and clean models. We trained 70 ResNet18 with clean MNIST dataset and another 70 ResNet18 using Trojaned MNIST dataset. Both groups of models have similar performance on clean testing images. Only Trojaned models will misclassify Trojaned images with high probability. Clean models will not be affected and will make correct prediction in spite of the trigger. Please refer to Sec. 5 for a more detailed description of the data generation procedure.

We extract topological features following the procedure introduced in the last section. As demonstrated in Fig. 2-(a), samples from the Trojaned dataset containing both clean and Trojaned examples are supplied to all the 140 networks. Neurons' activating values are recorded into a vector and the pairwise-correlation is calculated between all pairs of neurons. For each model, we build the simplicial complex, filter it based on the correlation, compute the persistence diagram, and extract topological features.

Please note that so far, to verify our hypothesis and to investigate its implication, we were using the *full-data* setting, i.e., using all training data to calculate neuron activation correlation. While this gives us the full picture of the network connectivity, and more reliable topological characterization, this is not a realistic setup for a Trojan detector. We will discuss how to extrapolate this to a more realistic *data-limited* setting in Sec. 5.

**Results.** Two topological features stand out, clearly differentiating Trojaned models and clean models: average death time of 0D homology class (connected components) and maximum persistence of the 1D homology class (cycles). As shown in Fig. 2-(b), the average deaths of connected components in Trojaned models are significantly smaller than those in clean models. The two-sample independent t-test is rejected at $99\%$ significance level. Note that here the filter function is one minus the correlation.

This implies that neurons in Trojaned models on average have larger correlation, and potentially tend to have larger cross-layer correlation. Possible explanation: the extra capacity is used in a Trojaned model to learn the trigger pattern, which causes more active neurons and consequently neurons are more likely to be correlated with each other through intermediate neurons.

Meanwhile, Fig. 2(c) shows significant topological signal in the maximum persistence of 1D homology. Intuitively, there exists a 1D cycle in Trojaned model that has significantly longer persistence than that in clean models (the two-sample independent t-test is rejected at 99% significance level). We inspect this phenomenon by identifying nodes and edges contained in the most significant cycle (Fig. 3). For a Trojaned model, the most significant cycle contains an edge linking a shallow layer and a deep layer. This is not the case for a clean model where a cross-layer edge is hardly ever spotted.

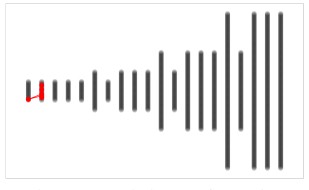

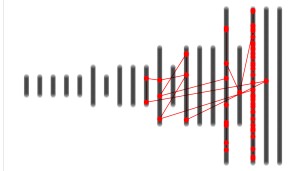

(a) Clean Model+Trojaned Input  (b) Trojaned Model+Trojaned Input

Figure 3: Most Persistent Cycles in ResNet18 with Death Time Cutoff at 0.35, on a clean (a) and a Trojaned model (b). On the Trojaned model, the loop consists of short cut connecting shallow and deep layers.

**Structural insight: the most persistent 1D cycle captures the short-cut.** We observe the high-persistence cycles of Trojaned models often contain strong-correlation edge connecting shallow and deep layers. We hypothesize that these cross-layer edges forms a **short cut** unique to Trojaned models. Neurons connected by the short-cut tend to fire together. This is sensible: Trojan triggers are often a localized pattern. They will be identified by shallow layer neurons. Meanwhile, for Trojaned network, the final prediction can be highly dependent on the identification of the trigger. Thus, there could be deep layer neurons (close to prediction) that are strongly connected to some shallow layer neurons (which activates when a Trigger is seen). See Figure 3 for illustrations.

### 4.3 Theoretical Guarantees

We conclude this section with a theoretical guarantee that the estimated persistence diagram will converge to a true one given sufficient samples. We prove the convergence in a population level.

Given $N$ potentially corrupted models $f_{1:N}$ and corresponding test input $X_{1:N}$, a natural practical concern about obtaining high quality approximation is the sample size requirement for each dataset $X_k, k \in [N]$. In particular, one needs to ensure that for all $N$ models the empirical estimation is faithful. A brief analysis shows we only need $O\left(\frac{\log(N)+\log(m)+\log(\frac{1}{\delta})}{\varepsilon^2}\right)$ samples as a minimum requirement for all $X_k$ to ensure that with high probability our empirically estimated persistence diagram $Dg(M(f, X), \mathcal{S})$ is sufficiently close to the ground truth $Dg(M(f, \mathcal{D}), \mathcal{S})$ in terms of bottleneck distance. We provide a proof in the supplemental material.

**Theorem 1.** *Let $M(f_k, X_k) \in \mathbb{R}^{m_k \times m_k}$ with $m_k \leq m^*, \forall k \in [N]$ and its entries $M_k^{i,j} = \frac{\Psi(v_i(X_k), v_j(X_k))}{\sqrt{\Psi(v_i(X_k), v_i(X_k))\Psi(v_j(X_k), v_j(X_k))}}$ and the its target value $M^*(f_k, \mathcal{D}_k) \in \mathbb{R}^{m_k \times m_k}$ with its entries $M_k^{*i,j} = \frac{\mathbb{E}_{X_k \sim \mathcal{D}_k}[\Psi(\boldsymbol{v}_i(X_k), \boldsymbol{v}_j(X_k))]}{\sqrt{\mathbb{E}_{X_k \sim \mathcal{D}_k}[\Psi(\boldsymbol{v}_i(X_k), \boldsymbol{v}_i(X_k))]\mathbb{E}_{X_k \sim \mathcal{D}_k}[\Psi(\boldsymbol{v}_j(X_k), \boldsymbol{v}_j(X_k))]}}$ as defined in section 3 with $\Psi(v_i(X), v_j(X)) = \frac{1}{n}\sum_{\boldsymbol{x}_l \in X} \psi(v_i(\boldsymbol{x}_l), v_j(\boldsymbol{x}_l))$. Suppose $\forall k \in [N], X_k$ are i.i.d. sampled from distribution $\mathcal{D}_k$ and $|\psi(v_i(\boldsymbol{x}), v_j(\boldsymbol{x}))| \leq \mathcal{R}$ for all $\boldsymbol{x} \sim \mathcal{D}_k, v_i, v_j, 0 < r \leq \mathbb{E}_{\boldsymbol{x} \sim \mathcal{D}_k}\psi(v_i(\boldsymbol{x}), v_i(\boldsymbol{x}))$ for all $i \in [m_k]$, if we have $\forall k \in [N]$,*

$$|X_k| \geq \frac{16\mathcal{R}^6\left(\log(N) + 2\log(m^*) + \log(\frac{1}{\delta})\right)}{r^4 \varepsilon^2}$$

*then with probability at least $1-\delta$, for all $k \in [N]$, $d_b(Dg(M(f_k, X_k), \mathcal{S}), Dg(M(f_k, \mathcal{D}_k), \mathcal{S})) \leq \varepsilon$.*

**Remark.** With the convergence theorem, it is not hard to show the following statement: Given sufficiently many samples, if we observe a gap in topology (persistent homology) between the estimated Trojaned and clean models, the gap likely also exists between the true models.

# 5   Application: A Topological Trojan Detector in Data-Limited Setting

In this section, we introduce an automatic Trojan detection algorithm based on our observation about Trojaned models' topological abnormality. The Trojan detection problem is essentially a classification problem. Given a set of training models, each clearly tagged as Trojaned or not, can we learn a classifier to predict whether a test model is Trojaned or not. Based on our previous study, we believe topological features can differentiate Trojaned models from clean ones. Our idea is to extract topological features from these models, and use them to train a classifier to predict the Trojan status of a test model. We have in total 12 topological features, including maximum persistence and average death (see supplemental for a complete list). We use a standard MLP (multi-linear perceptron) classifier. We will also provide the results using more powerful learning methods for persistence diagrams (e.g., using [6]) in the supplemental.

The major challenge is the limitation of data access. In practice, the Trojaned dataset will not be available to users. We adopt the data-limited setting: for each model (training or testing), only a few clean sample images are given. To acquire sufficient samples to estimate the correlation of each model, we apply a pixel-wise perturbation strategy. A formal algorithm of this is provided in the supplemental material. Given a clean sample image, we iterate through every pixel (or a small patch) and modify its value. Then such modified examples are all provided to the model as samples for building the correlation matrix.

To confirm that this sampling strategy is sufficient in mining the topological structure, we carry out the same hypothesis testing as in Sec. 4.2, except that we use the perturbed samples instead of the Trojaned dataset. As shown in Fig. 4, we still observe significant topological difference between the Trojaned and clean models. This gives us sufficient confidence to use topological features for Trojan detection, with the perturbed samples.

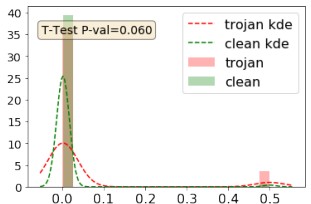
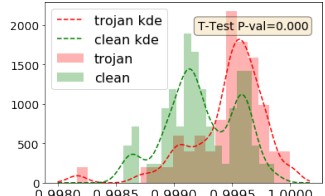

(a). Ave DeathTime - Perturbed Inputs        (b). Max-Persist - Perturbed Inputs

Figure 4: Ideal Feature Distribution v.s. Practical Feature Distribution. (a). Average death time calculated using real Trojaned data. (b). Average death time calculated using pixel-wise perturbed data.

Formally, we propose our Trojaned network detection algorithm in Alg. 1. We will release all our code for purpose of reproducibility. [4]. We validate our Trojan detector on synthetic and competition datasets, comparing with SoTA baselines.

**Synthetic Dataset Experiment.** We generate our synthetic dataset using NIST trojai toolkit[5]. In synthetic datasets, we trained 140 LeNet5 [40] and 120 ResNet18 [25] with MNIST [40] separately. We also trained 120 ResNet18 and 120 Densenet121 [31] with CIFAR10 [34] separately. Half of these models are trained with Trojaned datasets where we manually applied 20% one-to-one Trojan attack. Specifically, for Trojaned databases, we picked one of the source classes and added a reverse-lambda-shaped trigger (Figure 1) to a random corner of the input images. Then we changed the edited images' class to a predetermined target class and mixed them into the training database. Trojaned models are trained with these pollutant databases and clean models are trained with the original clean database. Furthermore, Trojaned models trained with MNIST datasets are constrained to maintain at least 95% successful attack rate (frequency of predicting the target class when trigger

---

[4]https://github.com/TopoXLab/TopoTrojDetection
[5]https://github.com/trojai/trojai

---

**Algorithm 1** Topological Abnormality Trojan Detection

---

1: **Input:** Training set models $\{f_1, f_2, \cdots, f_N\}$, Testing input associated with each model $X = \{X_1, X_2, \cdots, X_N\}$, Ground truth indicating Trojaned or not $Y = \{y_1, y_2, \cdots, y_N\}$
2: **Output:** Trojaned model detector $g$
3: **for** $i = 1, \cdots, N$ **do**
4: $\quad$ $X_i' = $ Pixel-wise Perturb $(X)$
5: $\quad$ Calculate correlation matrix $M(f_i, X_i')$
6: $\quad$ Build filtration of VR complex $\emptyset \subseteq \mathcal{S}_{t_1} \subseteq \mathcal{S}_{t_2} \subseteq \mathcal{S}_{t_T}$ using $M(f_i, X_i', \rho)$
7: $\quad$ Extract topological feature $z_i(\mathcal{S})$ as described in section 3
8: **end for**
9: Train Trojan detector $g$ with features $Z = \{\boldsymbol{z}_1, \boldsymbol{z}_2, \cdots, \boldsymbol{z}_{f_N}\}$ and Label $Y$

---

is presented on the test image) and models trained with CIFAR10 are constrained to maintain at least 87% successful attack rate. At the same time, MNIST models and CIFAR10 models also need to maintain at least 97% and 80% testing accuracy on clean inputs separately. There are no significant difference in terms of testing accuracy between clean models (on average 99% for MNIST and 84% for CIFAR10) and Trojaned models (on average 99% for MNIST and 84% for CIFAR10).

We also test our method with the recently proposed label consistent Trojan attack [57]. Such attack injects triggered input without modifying labels, thus can be potentially more stealthy. We follow the attack setting in the original paper. We synthetically inject 20% Trojan into target class 0 of CIFAR10. We use linear interpolation as the triggered sample generation method with hyper-parameter 0.3. We put trigger on all 4-corner on the image and make it invisible to blind eye as the original paper did. We choose the strongest attacking method to guarantee the attack successful rate. We train 54 clean Resnet32 and 57 Trojaned Resnet32. Clean models are required to have at least 75% test accuracy to be considered a valid data point. Trojaned models are required to have 75% on both clean and Trojaned examples to be considered valid. In a label-consistent Trojan Attack, even without target-class manipulation, a Trojaned model will predict a predetermined different class label on Triggered samples. The results on this attack is called "CIFAR10 Cons.+Resnet18".

We compare our Trojan detector's performance with several commonly cited approaches. (1) Neural cleanse (NC) [59], (2) Data-limited Trojaned network detection (DFTND) [62], (3) Universal litmus pattern (ULP) [32]. (4) Baseline classifier using correlation maxtrix directly (Corr). We evaluate using AUC (area under the curve) and ACC (accuracy). More experimental details are provided in supplemental material. Table 1 shows the results. We observe consistently that our method is superior compared with other baselines. Making highly accurate prediction of the Trojan status of test models. Our method outperforming baseline (4) shows that the short cut phenomenon cannot be directly captured by inspecting the correlation matrix. But our topological approach can capture it.

Table 1: Detection Performance on Synthetic Datasets

| ACC | | | | | |
|---|---|---|---|---|---|
| Dataset | NC | DFTND | ULP | Corr | Topo |
| MNIST+LeNet5 | $0.50 \pm 0.04$ | $0.55 \pm 0.04$ | $0.58 \pm 0.11$ | $0.59 \pm 0.10$ | $\mathbf{0.85 \pm 0.07}$ |
| MNIST+Resnet18 | $0.65 \pm 0.07$ | $0.53 \pm 0.07$ | $0.71 \pm 0.14$ | $0.56 \pm 0.08$ | $\mathbf{0.87 \pm 0.09}$ |
| CIFAR10+Resnet18 | $0.64 \pm 0.05$ | $0.51 \pm 0.10$ | $0.56 \pm 0.08$ | $0.72 \pm 0.07$ | $\mathbf{0.93 \pm 0.06}$ |
| CIFAR10+Densenet121 | $0.47 \pm 0.02$ | $0.59 \pm 0.07$ | $0.55 \pm 0.12$ | $0.58 \pm 0.07$ | $\mathbf{0.84 \pm 0.04}$ |
| CIFAR10 Cons.+Resnet18 | $0.55 \pm 0.11$ | $0.58 \pm 0.11$ | $0.78 \pm 0.04$ | $0.96 \pm 0.04$ | $\mathbf{1.00 \pm 0.00}$ |
| AUC | | | | | |
| Dataset | NC | DFTND | ULP | Corr | Topo |
| MNIST+LeNet5 | $0.48 \pm 0.03$ | $0.50 \pm 0.00$ | $0.54 \pm 0.12$ | $0.62 \pm 0.10$ | $\mathbf{0.89 \pm 0.04}$ |
| MNIST+Resnet18 | $0.64 \pm 0.11$ | $0.50 \pm 0.00$ | $0.71 \pm 0.14$ | $0.55 \pm 0.08$ | $\mathbf{0.97 \pm 0.02}$ |
| CIFAR10+Resnet18 | $0.63 \pm 0.06$ | $0.52 \pm 0.04$ | $0.55 \pm 0.05$ | $0.81 \pm 0.08$ | $\mathbf{0.97 \pm 0.02}$ |
| CIFAR10+Densenet121 | $0.58 \pm 0.12$ | $0.60 \pm 0.09$ | $0.52 \pm 0.02$ | $0.66 \pm 0.07$ | $\mathbf{0.93 \pm 0.03}$ |
| CIFAR10 Cons.+Resnet18 | $0.41 \pm 0.09$ | $0.50 \pm 0.00$ | $0.81 \pm 0.05$ | $1.00 \pm 0.00$ | $\mathbf{1.00 \pm 0.00}$ |

**Competition Dataset Experiment.** We also test our methods using IARPA/NIST trojai competition public dataset [50][6]. These datasets consist of synthetic traffic sign images superimposed on road background images. There are various model architectures available. In our experience, we mainly focus on ResNet and DenseNet. In this dataset, a randomly-generated polygon-shaped Trojan trigger (Figure 1-(a)) is overlaid on top of the foreground of $5\% \sim 50\%$ of training examples. The Trojaned model will predict the target class whenever a trigger is presented on the images for classes (all-to-one attack). All models have fixed 5 classes output. There are around 200 clean input images given as reference for each of these models.

For competition dataset, we let NC run with its default setting. To finish the experiment in a reasonable amount of time, we randomly pick 200 models from training to search for the optimal threshold for DFTND. For ULP, instead of looping through all models in every epoch, we randomly sampled a batch of 500 models for training. Following table shows the performance. Our method performs superior on this dataset.

Table 2: Detection Results on Competition Datasets.

| ACC | | | | |
|---|---|---|---|---|
| Dataset | NC | DFTND | ULP | Topo |
| ResNet | $0.63 \pm 0.03$ | $0.38 \pm 0.05$ | $0.63 \pm 0.00$ | $\mathbf{0.77 \pm 0.04}$ |
| DenseNet | $0.47 \pm 0.05$ | $0.49 \pm 0.04$ | $\mathbf{0.63 \pm 0.06}$ | $0.62 \pm 0.04$ |
| AUC | | | | |
| ResNet | $0.56 \pm 0.01$ | $0.45 \pm 0.05$ | $0.62 \pm 0.03$ | $\mathbf{0.87 \pm 0.03}$ |
| DenseNet | $0.42 \pm 0.03$ | $0.51 \pm 0.01$ | $0.63 \pm 0.06$ | $\mathbf{0.69 \pm 0.04}$ |

## 6  Conclusion

In this paper, we inspected the structure of Trojaned neural networks through a topological lens. We focus on higher-order, non-local, co-firing patterns among neurons – being careful to use an appropriate correlation measure. In particular, we observed – and statistically verified – the existence of robust topological structures differentiating between the Trojaned and clean networks. This revealed an interesting short-cut between shallow and deep layers of a Trojaned model. This topological methodology leads to a development of a competitive method of detecting Trojan attacks. More broadly, it appears this method could be adapted to other neural network structure analysis tasks – and perhaps promises ways of excising the undesirable structures.

**Acknowledgement**

The authors thank anonymous reviewers for their constructive feedback. The authors acknowledge support from US National Science Foundation (NSF) awards CRII-1755791, CCF-1910873, CCF-1855760 and IIS-1909038. This effort was partially supported by the Intelligence Advanced Research Projects Agency (IARPA) under the contract W911NF20C0038. The content of this paper does not necessarily reflect the position or the policy of the Government, and no official endorsement should be inferred.

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
