# Topological Detection of Trojaned Neural Networks
## – Supplementary Materials –

**Songzhu Zheng**[1]**, Yikai Zhang**[2]**, Hubert Wagner**[3]**, Mayank Goswami**[4]**, Chao Chen**[1]

[1]Stony Brook University, {zheng.songzhu,chao.chen.1}@stonybrook.edu
[2]Morgan Stanley, Yikai.Zhang@morganstanley.com
[3]University of Florida, hwagner@ufl.edu
[4]City University of New York, mayank.isi@gmail.com

In this supplemental material, we provide additional details on the theory, the algorithms, and the experiments. In Section 1, we provide a formal description of persistent homology, as well as the bottleneck distance. In Section 2, we continue analyzing the population level difference between Trojaned and clean models, with a focus on the short-cuts. Section 3 includes the proof of the proposition and theorem on (1) the existence of a topological discrepancy between Trojaned and clean models; (2) the convergence of the estimation of persistent homology in terms of the bottleneck distance. In Section 4, we provide more technical details on the experiments, including our sampling method based on pixel-wise perturbation, used baselines, and experiment configuration.

## 1 Persistent Homology and Bottleneck Distance

In the language of algebraic topology [9, 7], we can formulate a $p$-chain as a set of $p$-simplices. A boundary operator on a $p$-simplex takes all its adjacent $(p-1)$-simplices. In particular, the boundary of an edges consists of its adjacent nodes; the boundary of a triangle consists of its three edges. More generally, the boundary of a $p$-chain is the formal sum[1] of the boundary of all its elements, $\partial(c) = \sum_{\sigma \in c} \partial(c)$. Assume we have $m_p$ $p$-simplices for $p = 0, 1, 2$. If we fix an index of all the $p$-simplices, a $p$-chain uniquely corresponds to a $m_p$-dimensional binary vector. In dimension $p$, the boundary operator can be viewed as an $m_{p-1} \times m_p$ matrix, called the $p$-dimensional *boundary matrix*. It consists of the boundaries of all $p$-simplices, $\partial_p = [\partial(\sigma_1), \cdots, \partial(\sigma_{m_p})]$. It is often convenient to consider one big boundary matrix, whose blocks are the $p$-dimensional boundary matrices.

To compute persistent homology, we sort the rows and columns of the big boundary matrix according to the filter function values of the simplices. Then we apply a matrix reduction algorithm, similar to a Gaussian elimination – except we only allow left-to-right column additions. The classic algorithm [7] reduces the matrix from left to right, proceeding column by column. After the reduction, the pivoting entries of the reduced matrix correspond to pairs of simplices. We can interpret them as critical simplices that create and kill each topological structure. Their filter function values are the birth and death times of the corresponding persistence dot. The algorithm has worst-case cubic complexity, but modern implementations exhibits linear behaviour on practical inputs. This is an area of active research, and various algorithms were proposed to improve the algorithm either in theory [8, 4] or in practice [3, 2, 1]. In Figure 1, we show sample filtration complexes and the corresponding persistence diagram.

Aside from the birth-death pairs, we also add the set of all points on the diagonal line to the persistence diagram, i.e., $\mathrm{Dg}(M, \mathcal{S}) = \{(\mathrm{birth_i}, \mathrm{death_i})\} \cup \{(\mathrm{birth}, \mathrm{death}) | \mathrm{birth} = \mathrm{death}\}$.

---

[1]We remark that we focus on homology over the $\mathbb{Z}_2$ field, which is the simplest, but practical, setup. In this case the sums simply correspond to subsets of chains.

35th Conference on Neural Information Processing Systems (NeurIPS 2021), Sydney, Australia.

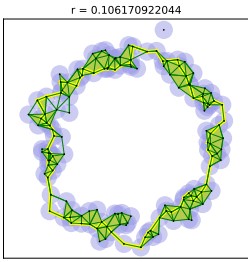
r = 0.106170922044

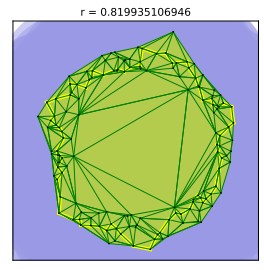
r = 0.819935106946

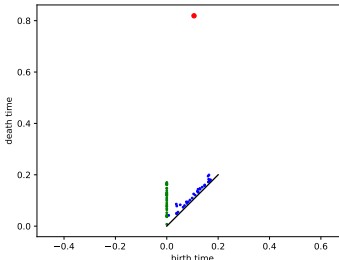

Figure 1: A finite set of points in $\mathbb{R}^2$ sampled with noise from an annulus. We see the union of Euclidean balls and the superimposed complex. Its vertices, edges and triangles depict the centers of the balls, pair-wise, and triple-wise intersections at two different radii. In our method we use the Vietoris–Rips complex, and here we only show its subcomplex to avoid visual clutter. The big loop indicated by the yellow closed curve is born on the *left* and dies on the *middle*. On the *right* we see the corresponding persistence diagram. The single dot in the upper part corresponds to the prominent feature, namely the big 1-dimensional cycle.

**Bottleneck distance.** We will use the bottleneck distance between two persistence diagrams [5]. Let $X$ and $Y$ be multisets of points corresponding to two diagrams we plan to compare. Let $\Gamma = \{\gamma : X \to Y\}$ be the family of bijections from $X$ to $Y$. The bottleneck distance is:

$$d_b(X, Y) = \inf_{\gamma \in \Gamma} \sup_{\boldsymbol{x} \in X} ||\boldsymbol{x} - \gamma(\boldsymbol{x})||_\infty.$$

It was shown that the bottleneck distance between diagrams is stable with regard to $L_\infty$ perturbation of the input filter function. Later on, Cohen-Steiner et al. [6] introduced the $p$-Wasserstein distance between diagrams and showed its stability, when assuming a Lipschitz condition of the filter function.

## 2 Statistical Inference Results on the Shortcut

In this section, we further investigate the existence of shortcuts through statistical analysis. In the main paper, we found that the Trojaned model can be identified through both 0D and 1D persistence diagrams, i.e., average death time of 0D diagram and maximum persistence of 1D diagram. Based on this, we hypothesize edges relevant to these topological features can be the shortcuts. For 0D, a shortcut could be an edge that kills connected component during the filtration. The filter function value $(1 - \rho_{i,j})$ of such edge is the death time. As shown in the main paper, the average death time of 0D diagrams clearly separates Trojaned and clean models. For 1D, a shortcut could also be the longest edge (i.e., the edge crossing the most layers of a neural network) in the high persistence 1D cycles.

We use persistent homology to select these shortcut candidates, and compare the length of these shortcut candidates from Trojaned models and clean models. Formally, we measure the *length* of an edge as the number of layers that an edge crossed (the index of layer contains the terminal neuron minus the index of layer contains the beginning neuron). For the purpose of verification, we use purely Trojaned examples to excite neurons and calculate the correlation matrix and VR filtration.

We first find all edges that kill a 0D homology class (called death edges) and measure their average length. The distribution of the average death edge length is displayed in Figure 2-(a). For each model, we only use the top 1000 edges (w.r.t. death time). Note many edges are connecting neurons within a same layer and have 0 length. As a result, their average length can be smaller than 1. We observe a significant difference between average lengths of the death edges of Trojaned and clean models. Trojaned models tend to have longer death edges. The two sample independent t-test average death edge length between Trojaned models and clean models is rejected at significance level smaller than $\leq 0.001$. Due to the computation precision reason, we round the p-value to the 4th digit. In reality, the significance level should be smaller than 0.001.

For 1D homology, we collect the longest edge in high persistence 1D homology cycles. For each model, we collect top 500 persistent 1D cycles. We extract the longest edges of these cycles and take

their average length. The distribution of the average length of these edges for different models is presented in Figure 2-(b). Similar to in 0D, the average length is generally low as we are averaging over many cycles. The distribution for clean models presents a bi-modal shape while the Trojaned models' is right skewed. On average, these type of edges in Trojaned models bypass more layers than those in clean models. The t-test's result corroborate our conclusion (p-value = 0.017), meaning a significance level of 95%.

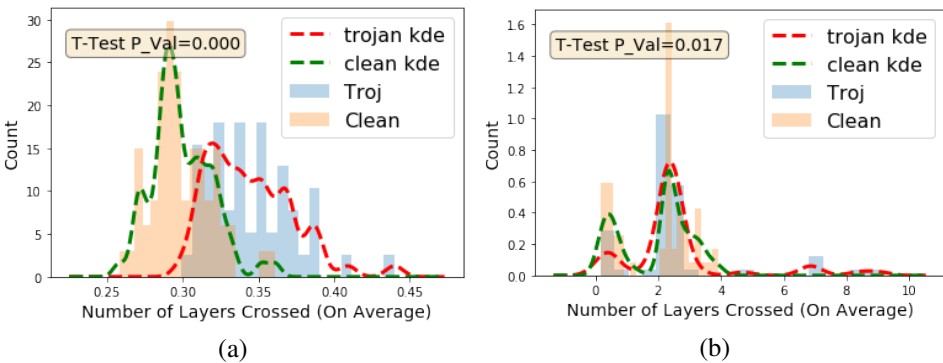

(a)                      (b)

Figure 2: Distribution of average lengths of shortcut edge candidates. (a). Average length of death edges in 0D persistence. Trojaned models generally have longer death edges. (b). Average length of the longest edges in high persistent 1D cycles. In Trojaned models, these edges are longer than in clean models.

We note that even though not all Trojaned models have significantly longer short cut edges than clean models, we do discover a significant subset of Trojaned models (about 10 out of 70) with long short cut edges from the top persistence cycles. We show a few samples in Figure 3. The behavior of these Trojaned models and their difference with the rest is yet to be further studied.

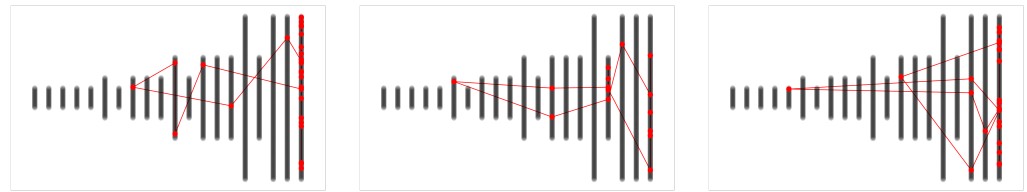

Figure 3: More examples of the top persistence cycles from Trojaned models.

# 3 Theorems and Proofs

In this section, we provide proof of the two theorems in the main paper.

## 3.1 Proof: Existence of Topological Discrepancy

Recall the definition of Trojaned Mix-Gaussian Triplet. An illustration can be found in Figure 4

**Definition 1** (Trojaned Mix-Gaussian Triplet). *Let* $\mu_1 = 2(-e_2 - e_1)\sigma\sqrt{\log(\frac{1}{\eta})}$, $\mu_2 = 2(-e_2 + e_1)\sigma\sqrt{\log(\frac{1}{\eta})}$, $\mu_3 = 2(e_2 - e_1)\sigma\sqrt{\log(\frac{1}{\eta})}$, $\mu_4 = 2(e_2 + e_1)\sigma\sqrt{\log(\frac{1}{\eta})}$. *Let* $i \sim uniform(\{1,2\})$ *and* $j \sim uniform(\{1,2,3,4\})$. *We define the following triple of distributions* $(\mathcal{D}_1, \mathcal{D}_2, \mathcal{D}_3)$ *to be Trojaned Mix-Gaussian Triplet (see supplementary section 1), where:*

$\mathcal{D}_1$*(Original data)* $= \{(\boldsymbol{x}, \boldsymbol{y}) : x \sim \mathcal{N}(\mu_i, \sigma^2 I_d), \ \boldsymbol{y} = i \ MOD \ 2\}$

$\mathcal{D}_2$*(Trojaned feature with correct labels)* $= \{(\boldsymbol{x}, \boldsymbol{y}) : x \sim \mathcal{N}(\mu_i, \sigma^2 I_d), \ \boldsymbol{y} = j \ MOD \ 2\}$

$\mathcal{D}_3$*(Trojaned feature with modified labels)* $= \{(\boldsymbol{x}, \boldsymbol{y}) : x \sim \mathcal{N}(\mu_i, \sigma^2 I_d), \ \boldsymbol{y} = \mathbb{1}_{j \in \{2,3\}}\}$

We study the hypothesis class $\mathcal{H}$ of binary output neural networks with two hidden layers and four neurons in each hidden layer equipped with an indicator activation function.

**Proposition 1.** *Let $(\mathcal{D}_1, \mathcal{D}_2, \mathcal{D}_3)$ be Trojaned Mix-Gaussian Triplet and $\mathcal{H}$ be the hypothesis class defined as above. Let $R(f, x, y) = \mathbb{1}(f(x) \neq y)$. There exists $f_1, f_2 \in \mathcal{H}$ where $\mathbb{E}_{(x,y)\sim\mathcal{D}_1}[R(f_1, x, y)] \leq \eta$, $\mathbb{E}_{(x,y)\sim\mathcal{D}_3}[R(f_2, x, y)] \leq \eta$, $\mathbb{E}_{(x,y)\sim\mathcal{D}_2}[R(f_2, x, y)] \geq \frac{1}{2}$, such that:*

$$d_b[Dg(M(f_1, \mathcal{D}_2), \mathcal{S}) - Dg(M(f_2, \mathcal{D}_2), \mathcal{S})] \geq 0.9$$

*where $d_b$ is bottleneck distance. $Dg(M(f_i, \mathcal{D}_2), \mathcal{S})$ is the 1D persistence diagram of the Vietoris–Rips filtration $\mathcal{S}$ that is built on top of the correlation matrix $M(f_i, \mathcal{D}_j)$.*

**Proof**: The proof is constructive. Let $f_1, f_2$ be parametrized by $U_1, V_1, W_1, b_1^U, b_1^V, b_1^W$ and $U_2, V_2, W_2, b_2^U, b_2^V, b_2^W$ and let

$$U_1 = \begin{bmatrix} e_1^\top \\ -e_1^\top \\ e_1^\top \\ -e_1^\top \end{bmatrix} \quad V_1 = \begin{bmatrix} 1 & 0 & 0 & 0 \\ 0 & 1 & 0 & 0 \\ 0 & 0 & 1 & 0 \\ 0 & 0 & 0 & 1 \end{bmatrix} W_1 = \begin{bmatrix} 0 & 1 \\ 1 & 0 \\ 0 & 1 \\ 1 & 0 \end{bmatrix}$$

$$b_1^U = \begin{bmatrix} 0 \\ 0 \\ 0 \\ 0 \end{bmatrix} \qquad b_1^V = \begin{bmatrix} -1 \\ -1 \\ -1 \\ -1 \end{bmatrix} b_1^W = \begin{bmatrix} 0 \\ 0 \\ 0 \\ 0 \end{bmatrix} \tag{1}$$

$$U_2 = \begin{bmatrix} e_1^\top \\ -e_1^\top \\ e_2^\top \\ -e_2^\top \end{bmatrix} \quad V_2 = \begin{bmatrix} 1 & 0 & 1 & 0 \\ 1 & 0 & 0 & 1 \\ 0 & 1 & 1 & 0 \\ 0 & 1 & 0 & 1 \end{bmatrix} W_2 = \begin{bmatrix} 0 & 1 \\ 1 & 0 \\ 0 & 1 \\ 1 & 0 \end{bmatrix}$$

$$b_2^U = \begin{bmatrix} 0 \\ 0 \\ 0 \\ 0 \end{bmatrix} \qquad b_2^V = \begin{bmatrix} -2 \\ -2 \\ -2 \\ -2 \end{bmatrix} b_2^W = \begin{bmatrix} 0 \\ 0 \\ 0 \\ 0 \end{bmatrix} \tag{2}$$

One can see $f_1(x) = \mathbb{1}(x^\top e_1 < 0)$ and $f_2(x) = \mathbb{1}(x^\top e_1 x^\top e_2 \geq 0)$ are Bayes optimal classifier for $\mathcal{D}_1$ and $\mathcal{D}_3$. Since $\|\mu_i - \mu_j\| \geq 4\sigma\sqrt{\log(\frac{1}{\eta})}$, the Bayes risk is at most $\eta$ which implies $\mathbb{E}_{(x,y)\sim\mathcal{D}_1}[R(f_1, x, y)] \leq \eta$ and $\mathbb{E}_{(x,y)\sim\mathcal{D}_3}[R(f_2, x, y)] \leq \eta$. If we use $\mathbb{1}(x^\top e_1 x^\top e_2 \geq 0)$ as decision boundary for classifying $(x, y)$ generated from $\mathcal{D}_2$, due its symmetricity exactly half of the samples will be misclassified thus $\mathbb{E}_{(x,y)\sim\mathcal{D}_2}[R(f_2, x, y)] \geq \frac{1}{2}$. Next we analyze the second moment matrix of neurons. Let $a_1 = \begin{bmatrix} p_1 \\ q_1 \end{bmatrix}$ and $a_2 = \begin{bmatrix} p_2 \\ q_2 \end{bmatrix}$, we next calculate $\mathbb{E}_{x\sim\mathcal{D}_1}[a_1 \otimes a_1]$ and $\mathbb{E}_{x\sim\mathcal{D}_1}[a_2 \otimes a_2]$.

$$\mathbb{E}_{x\sim\mathcal{D}_2}[p_1 \otimes p_1] = \begin{bmatrix} \frac{1}{2} & 0 & \frac{1}{2} & 0 \\ 0 & \frac{1}{2} & 0 & \frac{1}{2} \\ \frac{1}{2} & 0 & \frac{1}{2} & 0 \\ 0 & \frac{1}{2} & 0 & \frac{1}{2} \end{bmatrix} \mathbb{E}_{x\sim\mathcal{D}_2}[q_1 \otimes q_1] = \begin{bmatrix} \frac{1}{2} & 0 & \frac{1}{2} & 0 \\ 0 & \frac{1}{2} & 0 & \frac{1}{2} \\ \frac{1}{2} & 0 & \frac{1}{2} & 0 \\ 0 & \frac{1}{2} & 0 & \frac{1}{2} \end{bmatrix}$$

$$\mathbb{E}_{x\sim\mathcal{D}_2}[p_1 \otimes q_1] = \mathbb{E}_{x\sim\mathcal{D}_2}[(q_1 \otimes p_1)^\top] = \begin{bmatrix} \frac{1}{2} & 0 & \frac{1}{2} & 0 \\ 0 & \frac{1}{2} & 0 & \frac{1}{2} \\ \frac{1}{2} & 0 & \frac{1}{2} & 0 \\ 0 & \frac{1}{2} & 0 & \frac{1}{2} \end{bmatrix} \tag{3}$$

And $\mathbb{E}_{x \sim \mathcal{D}_2}[p_1] = [\frac{1}{2}, \frac{1}{2}, \frac{1}{2}, \frac{1}{2}]^\top$, $\mathbb{E}_{x \sim \mathcal{D}_2}[q_1] = [\frac{1}{2}, \frac{1}{2}, \frac{1}{2}, \frac{1}{2}]^\top$

$$\mathbb{E}_{x \sim \mathcal{D}_2}[p_2 \otimes p_2] = \begin{bmatrix} \frac{1}{2} & 0 & \frac{1}{4} & \frac{1}{4} \\ 0 & \frac{1}{2} & \frac{1}{4} & \frac{1}{4} \\ \frac{1}{4} & \frac{1}{4} & \frac{1}{2} & 0 \\ \frac{1}{4} & \frac{1}{4} & 0 & \frac{1}{2} \end{bmatrix} \quad \mathbb{E}_{x \sim \mathcal{D}_2}[q_2 \otimes q_2] = \begin{bmatrix} \frac{1}{4} & 0 & 0 & 0 \\ 0 & \frac{1}{4} & 0 & 0 \\ 0 & 0 & \frac{1}{4} & 0 \\ 0 & 0 & 0 & \frac{1}{4} \end{bmatrix}$$

$$\mathbb{E}_{x \sim \mathcal{D}_2}[p_2 \otimes q_2] = \mathbb{E}_{x \sim \mathcal{D}_2}[(q_2 \otimes p_2)^\top] = \begin{bmatrix} \frac{1}{4} & \frac{1}{4} & 0 & 0 \\ 0 & 0 & \frac{1}{4} & \frac{1}{4} \\ \frac{1}{4} & 0 & \frac{1}{4} & 0 \\ 0 & \frac{1}{4} & 0 & \frac{1}{4} \end{bmatrix}$$

(4)

And $\mathbb{E}_{x \sim \mathcal{D}_2}[p_2] = [\frac{1}{2}, \frac{1}{2}, \frac{1}{2}, \frac{1}{2}]^\top$, $\mathbb{E}_{x \sim \mathcal{D}_2}[q_2] = [\frac{1}{4}, \frac{1}{4}, \frac{1}{4}, \frac{1}{4}]^\top$

A simple calculation completes the proof. $\qquad\square$

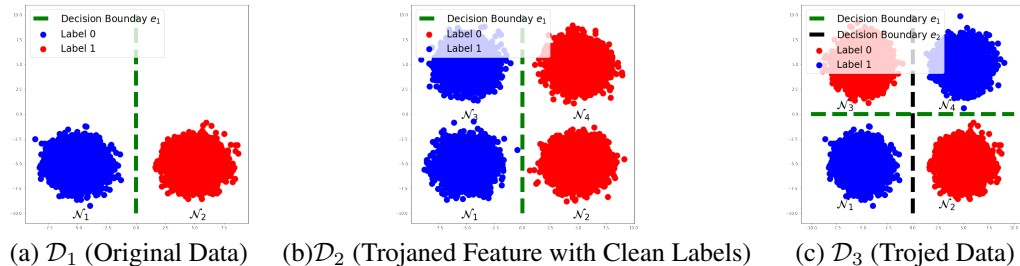

(a) $\mathcal{D}_1$ (Original Data)    (b)$\mathcal{D}_2$ (Trojaned Feature with Clean Labels)    (c) $\mathcal{D}_3$ (Trojed Data)

Figure 4: Demonstration of the Trojaned Gaussian Triplet. (a). $\mathcal{D}_1$ is the original data distribution. (b). $\mathcal{D}_2$ is the mixture data distribution of original distribution and the shifted distribution caused by trigger overlaying (note the classification risk is not necessary to be 0 to separate any two of these 4 Gaussian distribution). (c).$\mathcal{D}_3$ is the Trojaned dataset where labels will be modified for those Trojaned examples.

## 3.2 Proof: Convergence Theorem

**Theorem 1.** *Let* $M(f_k, X_k) \in \mathbb{R}^{m_k \times m_k}$ *with* $m_k \le m^*, \forall k \in [N]$ *and its entries* $M_k^{i,j} = \frac{\Psi(v_i(X_k), v_j(X_k))}{\sqrt{\Psi(v_i(X_k), v_i(X_k))\Psi(v_j(X_k), v_j(X_k))}}$ *and the its target value* $M^*(f_k, \mathcal{D}_k) \in \mathbb{R}^{m_k \times m_k}$ *with its entries* $M_k^{*i,j} = \frac{\mathbb{E}_{X_k \sim \mathcal{D}_k}[\Psi(\boldsymbol{v}_i(X_k), \boldsymbol{v}_j(X_k))]}{\sqrt{\mathbb{E}_{X_k \sim \mathcal{D}_k}[\Psi(\boldsymbol{v}_i(X_k), \boldsymbol{v}_i(X_k))]\mathbb{E}_{X_k \sim \mathcal{D}_k}[\Psi(\boldsymbol{v}_j(X_k), \boldsymbol{v}_j(X_k))]}}$ *as defined in section 3 with* $\Psi(v_i(X), v_j(X)) = \frac{1}{n}\sum_{\boldsymbol{x}_l \in X} \psi(v_i(\boldsymbol{x}_l), v_j(\boldsymbol{x}_l))$. *Suppose* $\forall k \in [N], X_k$ *are iid sampled from distribution* $\mathcal{D}_k$ *and* $|\psi(v_i(\boldsymbol{x}), v_j(\boldsymbol{x}))| \le \mathcal{R}$ *for all* $\boldsymbol{x} \sim \mathcal{D}_k, v_i, v_j, 0 < r \le \mathbb{E}_{\boldsymbol{x} \sim \mathcal{D}_k}\psi(v_i(\boldsymbol{x}), v_i(\boldsymbol{x}))$ *for all* $i \in [m_k]$, *if we have* $\forall k \in [N]$

$$|X_k| \ge \frac{16\mathcal{R}^6 \left(\log(N) + 2\log(m^*) + \log(\frac{1}{\delta})\right)}{r^4 \varepsilon^2}$$

*then with probability at least* $1 - \delta$, *for all* $k \in [N]$, $d_b(Dg(M(f_k, X_k), \mathcal{S}), Dg(M(f_k, \mathcal{D}_k), \mathcal{S})) \le \varepsilon$.

**Proof**: By Hoeffding inequality for each $\Psi(\boldsymbol{v}_i(X_k), \boldsymbol{v}_j(X_k))$, if $X_k$ has size $n_k \ge \frac{16\mathcal{R}^6\left(\log(N) + 2\log(m^*) + \log(\frac{1}{\delta})\right)}{r^4 \varepsilon^2}$ we have $|\Psi(\boldsymbol{v}_i(X_k), \boldsymbol{v}_j(X_k)) - \mathbb{E}_{X_k \sim \mathcal{D}_k}[\Psi(\boldsymbol{v}_i(X_k), \boldsymbol{v}_j(X_k))]| \le \frac{\varepsilon r^2}{4\mathcal{R}^2}$ with probability at least $1 - \frac{\delta}{m^{*2}N}$.

Next we bound

$$\left| \frac{\Psi(\boldsymbol{v}_i(X_k), \boldsymbol{v}_j(X_k))}{\sqrt{\Psi(\boldsymbol{v}_i(X_k), \boldsymbol{v}_i(X_k))\Psi(\boldsymbol{v}_j(X_k), \boldsymbol{v}_j(X_k))}} - \frac{\mathbb{E}\left[\Psi(\boldsymbol{v}_i(X_k), \boldsymbol{v}_j(X_k))\right]}{\sqrt{\mathbb{E}\left[\Psi(\boldsymbol{v}_i(X_k), \boldsymbol{v}_i(X_k))\right]\mathbb{E}\left[\Psi(\boldsymbol{v}_j(X_k), \boldsymbol{v}_j(X_k))\right]}} \right|$$

(5)

By setting $a_1 = \Psi(\boldsymbol{v}_i(X_k), \boldsymbol{v}_j(X_k)), a_2 = \sqrt{\mathbb{E}\left[\Psi(\boldsymbol{v}_i(X_k), \boldsymbol{v}_j(X_k))\right]}, b_1 = \sqrt{\Psi(\boldsymbol{v}_i(X_k), \boldsymbol{v}_i(X_k))}, b_2 = \mathbb{E}\left[\Psi(\boldsymbol{v}_i(X_k), \boldsymbol{v}_i(X_k))\right], c_1 = \sqrt{\Psi(\boldsymbol{v}_j(X_k), \boldsymbol{v}_j(X_k))}, c_2 = $

$\sqrt{\mathbb{E}\left[\Psi(\boldsymbol{v}_j(X_k), \boldsymbol{v}_j(X_k))\right]}$, we can observe that $\frac{a_1}{b_1 c_1} - \frac{a_2}{b_2 c_2} = \frac{a_1 b_2 c_2 - a_2 b_1 c_1}{b_1 c_1 b_2 c_2}$. Due to the fact that $|a_1 - a_2| \leq \frac{\varepsilon r^2}{4\mathcal{R}^2}, |b_1^2 - b_2^2| \leq \frac{\varepsilon r^2}{4\mathcal{R}^2}, |c_1^2 - c_2^2| \leq \frac{\varepsilon r^2}{4\mathcal{R}^2}$ and $a_1 \leq \mathcal{R}, a_2 \leq \mathcal{R}, r \leq b_2^2 \leq \mathcal{R}, r \leq c_2^2 \leq \mathcal{R}$, we have $b_1 c_1 b_2 c_2 \geq \frac{r^2}{4}$ and $|a_1 b_2 c_2 - a_2 b_1 c_1| \leq 2\varepsilon \mathcal{R}^2$ which implies that Equation (5) is bounded by $\varepsilon$. By taking a union bound on failure probability for all $m_k^2$ entries in matrix $M_k$ and for all $M_k, k \in [N]$ one will get with probability at least $1 - \delta$:

$$\forall k \in [N], \|M(f_k, X_k) - M(f_k, \mathcal{D}_k)\|_\infty \leq \varepsilon$$

By stability theorem of bottleneck distance [5] with probability at least $1 - \delta$ for all $k \in [N]$:

$$d_b(\text{Dg}(M(f_k, X_k), \mathcal{S}), \text{Dg}(M(f_k, \mathcal{D}_k), \mathcal{S})) \quad \leq \|M(f_k, X_k) - M(f_k, \mathcal{D}_k)\|_\infty \leq \varepsilon$$

$\square$

## 4 Experimental Details

### 4.1 Pixel-wise Perturbation

For Trojan detection, we are only given a few clean samples for each model. We propose a pixel-wise perturbation algorithm to obtain samples. See Algorithm 1.

---
**Algorithm 1** Pixel-wise Perturbation
---
1: **Input:** Dataset $X = \{\boldsymbol{x}_1, \boldsymbol{x}_2, \cdots, \boldsymbol{x}_m\}$, Number of trials $n$, Input Range $L = \{(\boldsymbol{l}_1, \boldsymbol{u}_1), (\boldsymbol{l}_2, \boldsymbol{u}_2), \cdots, (\boldsymbol{l}_m, \boldsymbol{u}_m)\}$
2: **Output:** Coordinate Perturbed Dataset $X'$
3: $X' = \emptyset$
4: **for** $i = 1, \cdots, m$ **do**
5: $\quad$ $X'_i = \emptyset$
6: $\quad$ **while** $j \leq n$ **do**
7: $\quad\quad$ $\boldsymbol{x}_i^c = \boldsymbol{x}_i$
8: $\quad\quad$ Sample $k \sim \{1, 2, \cdots, d\}$, sample a perturbed value $v \sim [\boldsymbol{l}_i, \boldsymbol{u}_i]$
9: $\quad\quad$ Set $k$th coordinate of $\boldsymbol{x}_i^c[k] = v$
10: $\quad\quad$ $X'_i = X'_i \cup \boldsymbol{x}_i^c$
11: $\quad\quad$ $j + +$
12: $\quad$ **end while**
13: $\quad$ $X' = X' \cup X'_i$
14: **end for**
15: **Return:** $X'$
---

### 4.2 Synthetic Experiment Baseline Setting and Experiment Configuration

**Baseline Setting.** We compare our Trojan detector's performance with several commonly cited approaches. Neural cleanse (NC) introduces a reversed engineering approaches where the algorithm tries to find a pattern when overlaying with input can flip the output of the model. It detects a Trojaned model if the median absolute deviation of any resulting reverse engineered pattern goes beyond 2. Data-limited Trojaned network detection (DFTND) identifies a Trojaned model if the difference between the norm of the penultimate layer's representation of a clean input and a adversarial input goes above certain threshold. Universal litmus pattern (ULP) adopts a meta training idea where several randomly initialized examples (ULP) are given to all models. These ULPs are optimized to form representations that can be learned by a Trojan detector to discriminate clean and Trojaned models. We also compare with a baseline classifier that exploits the correlation matrix directly (Corr). We extract the top 5 singular values of the correlation matrix and calculate the Frobinius norm of the matrix after thresholding using $25\%, 50\%, 75\%$ percentile of the matrix separately. We combine these values into a feature vector and train a classifier with these feature.

**Experiment Configuration.** We use $80\%$ of the data as the training set and use the rest $20\%$ as the testing set. NC doesn't need training set so we randomly choose $20\%$ of data to measure the performance. DFTND doesn't require training set either. So we use the training set to search for a

optimal threshold that minimize the cross entropy loss on training set. We repeat each experiment 5 times and the results are record in Table 1 and Table 2 in the main text. Our detector's performance is consistently better than all baselines.