# OpenReview forum: "Topological Detection of Trojaned Neural Networks"
_NeurIPS.cc/2021/Conference — NeurIPS 2021 Poster_

### Official Review · Reviewer_3K3q · 2021-07-15

**Rating:** 7
**Confidence:** 3

**Summary:**

This paper presents an approach for the detection of malicious backdoored neural networks using persistent homology from topological data analysis.
First, a neuron connectivity graph is built.
This is a complete weighted graph whose nodes correspond to different neurons of a neural network.
Each edge connects two different neurons, and its weight is set to be negatively proportional to the correlation of the activation vectors of the two neurons evaluated for the input data.
Having this graph, its topological features are extracted using the Vietoris-Rips filtration.
Using this framework, it is then argued (both theoretically and empirically) that there exist topological dissimilarities between clean and poisoned models.
Based on this topological analysis, it is shown that in Trojaned models: (1) the neurons exhibit a larger correlation with each other, and (2) there exists a short-cut path connecting shallow and deep layers.
Further, a theoretical guarantee is provided to show the convergence of the estimated topological features using sufficient data samples to the true one.
Finally, a practical algorithm for backdoor model detection is proposed.
The experiments indicate the better performance of the proposed approach compared to well-known baselines such as neural cleanse [1], universal litmus patterns [2], and data-limited Trojaned network detection [3].

**Limitations And Societal Impact:**

The potential negative societal impacts are not discussed.
Perhaps you should discuss how the method developed here can be exploited by an adversary.

**Main Review:**

### Strengths and Weaknesses:
+ First, the paper is very well-written.
Considering the novelty of persistent homology to the machine and deep learning audience, this paper does an extremely good job of describing such concepts in a straightforward, yet informative, way.
Nonetheless, it would benefit from careful proofreading. See the additional feedback for a few minor suggestions.

+ More importantly, the contributions of this paper are sound and promising.
The proposed approach sets up a novel direction in the analysis of Trojaned neural networks using topological data analysis.
These findings are analyzed meticulously, shedding light on the behavior of neural networks under backdoor attacks.
Although that some of these behaviors, such as the existence of a short-cut in backdoored models, have been previously discussed (see [4] for e.g.), this paper uses a different, novel route to reach the same conclusions.

+ The empirical results indicate the potential of the proposed approach in the detection of backdoored neural networks in contrast to the state-of-the-art.

+ The main concerns with the current version are three-fold:

	1. The relevance of the theoretical study in Section 4.1 to Trojan attacks is obscure.
	Specifically, the data that is used in this study do not resemble the general settings in backdoor attacks:

		- number of poisoned and clean data are equal, and
		- the final poisoned dataset $\mathcal{D}_3$ is completely different from the clean data distribution, and one cannot expect that training a model on it can be equal to training on the clean data.

	Perhaps a point is missing here, and this should be made clear in Section 4.1.

	2. Throughout the paper, especially for the empirical studies in Section 4.2 which form the foundation of the proposed approach, it is assumed that the data poisonings are done in a dirty-label fashion, meaning that the attacker flips the labels besides injecting triggers.
	In the past couple of years, there has been a good progress in stealthy backdoor attacks that do not change the labels, e.g., see [5].
	To make more conclusive judgements about Trojaned models, such attacks should perhaps be considered.

	3. Possible shortcomings of the proposed approach are not discussed.
	For instance, does the current approach only work for the detection of networks with the same architecture?
	In other words, can one train the detector on say ResNet-18, and detect malicious DenseNets?
	Furthermore, does the poisoned models used for training and testing use different triggers?
	Or the training and test models should be poisoned with the same triggers?
	These are key questions when it comes to backdoor model detection and should be shown through experiments.

### Additional Feedback:
+ It would be beneficial if the Gebhart and Schrater [6] paper regarding the detection of adversarial attacks in neural networks using persistent homology is added to the list of related work.

+ Line 108 says "These Trojaned samples are assigned specific target class labels – different from the labels of the original training samples."
As stated before, this is not always the case.
Consider adding a few sentences to state that this is the assumption in the paper.

+ In the Vietoris-Rips filtration, why $t$ needs to be varied from $-\infty$ to $\infty$?
Because the edge weights are always between -1 and 1.

+ In Definition 1, datasets $ \mathcal{D}_2 $ and $ \mathcal{D}_3 $ should use $\mu_j$ instead of $\mu_i$.

+ Consider adding different hatches and line-styles to Figures 2 and 4 to help with their readability for the color-blind.

+ In Figure 4, apparently some of the sub-figures are missing since the caption has 4 parts but there are only two sub-figures.

+ In Algorithm 1, does one have to use a different testing for each poisoned model?
Or the same one can also be used multiple times?

+ Minor suggestions, typos, and grammatical errors:
	1. Line 3 and 4: do not split the paragraphs in the abstract.
	2. Line 15: "attach" -> "attack"
	3. Line 75: add a few references for the sentence "Few methods investigate..."
	4. Line 116: "Let clean dataset $\mathcal{D}$..." -> "Let the clean dataset be $\mathcal{D}$..."
	5. Theorem 2: "iid" -> "i.i.d."
	6. Line 286: "can we" -> "we can"
	7. Line 297: "is provide" -> "is provided"
	8. Line 298 and 299: "(or a small patch) modify its value." -> "(or a small patch) and modify its value."
	9. Line 349 and 350: "These topological methodology lead to..." -> "This topological methodology leads to..."
	10. Line 155: "is a collections of..." -> "is a collection of..."

#### References

[1] Wang, Bolun, et al. "Neural Cleanse: Identifying and mitigating backdoor attacks in neural networks." _IEEE Symposium on Security and Privacy (SP)_, 2019.

[2] Kolouri, Soheil, et al. "Universal litmus patterns: Revealing backdoor attacks in CNNs." _CVPR_, 2020.

[3] Wang, Ren, et al. "Practical detection of Trojan neural networks: Data-limited and data-free cases." _ECCV_, 2020.

[4] Li, Shaofeng, et al. "Invisible backdoor attacks on deep neural networks via steganography and regularization." _IEEE Transactions on Dependable and Secure Computing_ (2020).

[5] Turner, Alexander, Dimitris Tsipras, and Aleksander Madry. "Label-consistent backdoor attacks." _arXiv preprint arXiv:1912.02771_ (2019).

[6] Gebhart, Thomas, and Paul Schrater. "Adversary detection in neural networks via persistent homology." _arXiv preprint arXiv:1711.10056_ (2017).

**Time Spent Reviewing:**

5:30

---

> ### Author Response · Authors · 2021-08-10
> **Response to Reviewer 3K3q**
>
> We are glad the reviewer appreciates the novelty, the thorough empirical evaluation, the theoretical soundness and the carefully tuned presentation of the paper. We will include and carefully discuss the new citations pointed out by the reviewer:
>
> [4] Li, Shaofeng, et al. "Invisible backdoor attacks on deep neural networks via steganography and regularization." IEEE Transactions on Dependable and Secure Computing (2020).
>
> [5] Turner, Alexander, Dimitris Tsipras, and Aleksander Madry. "Label-consistent backdoor attacks." arXiv preprint arXiv:1912.02771 (2019).
>
> [6] Gebhart, Thomas, and Paul Schrater. "Adversary detection in neural networks via persistent homology." arXiv preprint arXiv:1711.10056 (2017).
>
> Meanwhile, we will carefully proofread the paper and fix the typos and grammatical errors pointed out by the reviewer.  Below we address more specific questions one-by-one.
>
> __Q1.__ The relevance of the theoretical study in Section 4.1 to Trojan attacks is obscure. Specifically, the data that is used in this study do not resemble the general settings in backdoor attacks.
>
> __Answer:__ Thank you for pointing this out. We agree that the constructed example is too specific and may not represent the real world scenario. As we stated in the overall response, the goal of this constructive example is merely to demonstrate the existence of a Trojaned distribution with a mutated network topology. We will clarify this in the paper.
>
> __Q2.__ This paper assumed that the data poisonings are done in a dirty-label fashion, meaning that the attacker flips the labels besides injecting triggers. Should perhaps consider stealthy backdoor attacks that do not change the labels, e.g., see [5].
>
> __Answer:__ Thank you for the excellent suggestion. This would be an interesting future topic to investigate such types of attack. We will discuss this in the paper.
>
> __Q3.__ Possible shortcomings of the proposed approach are not discussed.
>
> __Answer:__ One assumption is that our method assumes relatively localized triggers that activate shallow layer neurons (low level image features). It will be interesting to investigate whether the algorithm extends to other more global triggles (e.g., color filter). Also, as you have pointed out, our method mainly deals with label-corruption attacks with local triggers. We haven’t investigated its detection power on label consistent attacks.
>
> __Q4.__ Does the current approach only work for the detection of networks with the same architecture? In other words, can one train the detector on say ResNet-18, and detect malicious DenseNets? Furthermore, does the poisoned models used for training and testing use different triggers? Or the training and test models should be poisoned with the same triggers? These are key questions when it comes to backdoor model detection and should be shown through experiments.
>
> __Answer:__ Excellent questions. Below we demonstrate quantitatively that the proposed method does transfer across model architectures and across different triggers. We hypothesize that this strength is due to the fact that the topological features are global structural characteristics that are robust to variations of models and triggers.
>
> __Transferring across network architectures__. We train a topological trojan detector using models of one specific architecture (source architecture) and test its detection performance on models of another architecture (target architecture).
>
> | Dataset | Source | Target | ACC | AUC |
> |---|---|---|---|---|
> | MNIST | Resnet18 | Lenet5 | $0.70 \pm 0.10$ | $0.76 \pm 0.07$ |
> | MNIST | Lenet5 | Resnet18 | $0.80 \pm 0.07$ | $0.95 \pm 0.04$ |
> | CIFAR10 | Resnet18 | Densenet121 | $0.63 \pm 0.06$ | $0.74 \pm 0.07$ |
> | CIFAR10 | Densenet121 | Resnet18 | $0.63 \pm 0.00$ | $0.61 \pm 0.07$ |
>
> We observe that after transferring, the method still maintains a decent performance. It has equal or better performance than all baselines that are trained using the target architecture (Tables 1 in the main paper).
>
> __Transfer across Trigger shape__. We use NIST/ARPAR trojai competition round1 dataset and focus on ResNet architecture. The triggers in this dataset can be of different sizes. We use triggers of 10% image size (small triggers) and triggers of 50% image size (large triggers). We train the topological trojan detector using models that are attacked with triggers of one size (source) and test its detection ability on models trojaned by triggers of another size.
>
> | Source | Target | ACC | AUC |
> |:---|:---:|:---:|:---:|
> | 10\% | 50\% | $0.66 \pm 0.02$ | $0.83 \pm 0.02$ |
> | 50\% | 10\% | $0.85 \pm 0.05$ | $0.95 \pm 0.00$ |
>
> We observe that after transferring, the method still maintains a decent performance.
>
> __Q5.__ It would be beneficial if the Gebhart and Schrater [6] paper regarding the detection of adversarial attacks in neural networks using persistent homology is added to the list of related work.
>
> __Answer:__ Thanks for pointing out this paper. It is very interesting and relevant. Will definitely cite and discuss in the paper.
>
> __Q6.__ Line 108 says "These Trojaned samples are assigned specific target class labels – different from the labels of the original training samples." As stated before, this is not always the case. Consider adding a few sentences to state that this is the assumption in the paper.
>
> __Answer:__ Yes. Will clarify this assumption and discuss other types of attacks, e.g., [5].
>
> __Q7.__ In the Vietoris-Rips filtration, why t needs to be varied from $−\infty$ to $\infty$ Because the edge weights are always between -1 and 1.
>
> __Answer:__ Yes, in our work it is sufficient to vary $t$ between -1 and 1. We were writing it in a general context, in which $t$ can be arbitrary. We will clarify this.
>
> __Q8.__ In Definition 1, datasets $D_2$ and $D_3$ should use $\mu_j$ instead of  $\mu_i$
>
> __Answer:__ Thanks for pointing out the typo. Will fix.
>
> __Q9.__ Improving Fig 2 and Fig 4.
>
> __Answer:__ Thank you for these suggestions. Will improve accordingly.
>
> __Q10.__ In Algorithm 1, does one have to use a different testing for each poisoned model? Or the same one can also be used multiple times?
>
> __Answer:__ We assume you meant “testing image”. We use examples from the dataset on which the poisoned model is trained on. However, in practice, we have observed that randomly initialized images also work well. These random images can be used across models.
>
> __Q11.__ Minor suggestions, typos, and grammatical errors:
>
> __Answer:__ Thanks for these detailed comments. Will improve our paper accordingly.

---

> > ### Comment · Reviewer_3K3q · 2021-09-01
> > **Thanks for your response**
> >
> > Dear authors,
> >
> > Thanks for your response.
> >
> > The new experimental results on the transferability of the proposed detection method seem encouraging.
> > It would be beneficial if this new transferability experiment as well as a comparison with your baselines are added to the revised paper.
> > Also, as mentioned in my main review, the current method is only applied to dirty-label backdoor attacks, while many recently introduced attacks work without poisoning the labels (the so-called clean-label attacks).
> > To provide a clearer picture of the proposed method's performance, some experiments on this type of attack need to be added to the paper.
> >
> > Overall, the rebuttal has addressed some of my concerns.
> > Also, I like the proposed idea of using persistent homology to explain backdoor attacks in neural networks, and I think the backdoor attacks' community can be inspired by this novel route.
> > As such, I decided to increase my score to 7.
> >
> > Sincerely.

---

> > > ### Author Response · Authors · 2021-09-02
> > > **Further Reply to Reviewer 3K3q**
> > >
> > > We would like to thank the reviewer for the extra response -- and for all the constructive comments. We will of course follow the suggestions and include the existing transferability experiments as well as add new experiments on the clean-label attacks -- thank you for pointing that out.
> > >
> > > Again: thank you for the efforts to improve our work and for your support.

---

> ### Author Response · Authors · 2021-08-30
> **Follow Up on Our Initial Response**
>
> We really appreciate Reviewer 3K3q’s thorough and constructive initial review. We are wondering if our initial response has addressed the concerns to a satisfactory degree. The question about the transferability of the proposed method lead to new experimental results. We would appreciate any further comments and suggestions to improve the paper.

---

### Official Review · Reviewer_r1eN · 2021-07-20

**Rating:** 6
**Confidence:** 3

**Summary:**

The authors investigate an approach to detecting Trojan Networks, using topological features based on the correlation matrices of the network activations.  They provide primarily empirical evidence that such features are discriminative for Trojan networks, including a comparison of the method against previous approaches on a synthetically designed task using MNIST and CIFAR10 datasets.  The further provide a theoretical analysis on a toy problems (Th. 1) and provide an analysis of the estimation error of the persistence diagram necessary for their approach (Th. 2).

**Ethical Concerns:**

Not relevant.

**Limitations And Societal Impact:**

Not relevant.

**Main Review:**

The results are interesting, and provide a compelling case for the usefulness of topological features in detecting Trojan networks.  This is the primary contribution of the paper, which appears to be a robust phenomenon in the contexts they test.  The theoretical analysis of the toy problem in Def. 1 and Th. 1 is less convincing, since this appears to establish only that in this setting, topologically distinct models can be found for the clean/Trojan settings, and not that this must or is more likely to be the case (indeed, the same may be true for any distinct conditions, say with the output labels grouped into meta-classes).  Also, the definition of what counts as a trigger (Sec 2) appears subjective, although this is not specific to the paper.  However, the paper is well written and clearly presented, appears to be theoretically and empirically sound in terms of the claims made, and the phenomenon observed is likely to be of interest, and may lead to further work on theoretical understanding.

Minor points:

The paper should be checked for typos before the final version.

**Time Spent Reviewing:**

2 hours

---

> ### Author Response · Authors · 2021-08-10
> **Response to Reviewer r1eN**
>
> We are glad that the reviewer appreciated the presentation, the sound theoretical contribution and empirical observation. We are particularly excited that the reviewer noticed the great theoretical potential of this work (see Q3 below for discussions). Below we address specific questions.
>
> __Q1.__ Concerns regarding Def 1. and Th. 1.
>
> __Answer:__ Please refer to our overall response for our answer to this question.
>
> __Q2.__ The definition of a trigger (sec 2) is subjective.
>
> __Answer:__ Thank you for pointing this out. Yes. We are assuming that a trigger is a certain localized pattern overlaid on the image. We will make it explicit in the paper. It will be an interesting future work to explore how our analysis can be extended to other contexts like classification with hard examples or adversarial attacking.
>
> __Q3.__ “[...] the phenomenon observed is likely to be of interest, and may lead to further work on theoretical understanding.”
>
> __Answer:__ Thank you for pointing this out. Yes! It is evident that many interesting questions open up. To name a few: Can a similar methodology help understand neural networks also outside of the Trojan attack context? What kind of topologies do networks generate? How is it affected by training datasets and how does the topology affect the network’s performance, ability to generalize etc? More on the TDA part: Can we understand which details make our particular topological construction work? In particular, can we improve it by switching the inherently approximative VR filtration, to something more exact like a Cech filtration? Can we say something about the geometry of the space whose topology we approximate (perhaps making a connection with manifold learning)?
>
> Finally, we mention that -- as a side-product of this work -- we now have computational tools that can help answer this kind of questions.
>
> __Q4.__ check for typos.
>
> __Answer:__ thank you. We will carefully proofread the paper.

---

### Official Review · Reviewer_wnSc · 2021-08-02

**Rating:** 6
**Confidence:** 2

**Summary:**

This submission considers how to detect Trojaned Neural Networks from a particular feature extraction method - which the authors call topological descriptor. Overall, my impression is that the paper neglects the duty of defining the problem in a concise, mathematical language - I still don't understand what the proposed algorithm does to extract features, descriptions are too handwavy.
If assuming some black-box method is given to extract (topological) features of fully-trained neural network, main claim is that it is easy to classify whether a (fully-trained) Neural-Net is Trojaned or not, a reduced-task which I believe also should be formally defined. Empirically, extracted features are shown to be well-separated between Trojaned and Non-Trojaned neural networks.

**Limitations And Societal Impact:**

See above.

**Main Review:**

Writing

There are too many places where descriptions should be more concise and precise. Here are a few examples

1. Line 116-122: what is "m" and "x"? Are they vectors or matrices? How are (1-m) and multiplication defined?

2. Line 146-153: what is, and how do you compute $\rho$? What are neurons here? Please define it mathematically.

3. Line 168-177: on birth/death time: where the notion of "time" comes from? In which procedure?

There are also too many jargons which are not very standard in ML community, like "persistence diagrams" or "cycles of high persistence" in Line 184-185. They are not defined anywhere in the paper. Overall the paper is very hard to read and follow.




- Theory

Theories do not back-up the idea anyway either. For instance,

Theorem 4.1: a cherry-picked good example (Gaussian ones here) does not justify your method. It may be used to give a good intuition, but it doesn't go the other way - your method should be general enough to handle broad class of examples.

Theorem 4.2: This is not much more than a "uniform concentration bound" for a finite function-class. And what is $\psi$? is it ever defined before?


- Experiment

I don't understand why your experiment should suddenly be in data-limited setting, while your theory is based on the law of large numbers. Numbers in Table 1 seems impressive, probably this is the strength of the paper.



Overall, I think the paper is poorly written and should be significantly revised, but there seems to be potential as an empirical work to detect Trojaned Neural Networks. At this time, I recommend rejection.


**Time Spent Reviewing:**

4

---

> ### Author Response · Authors · 2021-08-10
> **Response to Reviewer wnSc**
>
> We appreciate the reviewer’s opinion. Our paper is at the intersection of machine learning and the advanced theory of topological data analysis. To make it accessible to most of the NeurIPS audience, we took special care to introduce concepts from this theory in an intuitive way -- the original algebraic-topological definitions would be lengthy and opaque for most readers. Our strategy seems successful as both of the other reviewers commended our paper as “well written”.
>
> We’d like to emphasize that our topological descriptor is built on a well-established mathematical theory of persistent homology [4,6,12,17,18]. The innovation of this paper is the application of such theory to the analysis and detection of Trojan-attacked neural networks.
>
> We do agree that defining the crucial concepts in a more rigorous way will strengthen the paper. Below, we will carefully address specific questions raised by the reviewer. We will incorporate these answers into the main paper. We also plan to further expand the supplemental material with a more formal background of the theory; it will complement the main paper well, especially for readers interested in the theory.
>
> ### Writting
>
> __Q1.__ Line 116-122: what is "$m$" and "$x$"? Are they vectors or matrices? How are ($1-m$) and multiplication defined?
>
> __Answer:__  We will clarify in the paper that $m$ is a binary mask matrix with the same shape as the input image $x$. To mask an image, we use elementwise product operation (called the Hadamard product in the literature). Also ($1-m$) is the element-wise subtraction operation, where 1 is implicitly an all-ones matrix of the same shape as $m$ and $x$. These are standard conventions in image processing, and in particular we follow the definitions in [57, 59].
>
> __Q2.__ Line 146-153: what is, and how do you compute ρ ? What are neurons here? Please define it mathematically.
>
> __Answer:__ For a standard neural network with input $x$, the $k$’s neuron at layer $l$ is
> $k$th element of $\sigma(W^l ( \cdots \sigma (W^1 x)))$ , in which $\sigma$ is the nonlinear activation function, e.g., ReLU. $W^l$ is an $n_{l}\times n_{l-1}$ weight matrix between layer $l-1$ and layer $l$, with $n_{l-1}$ and $n_l$ neurons respectively. We collect all the neurons through different layers and index them from 1 to $m$.
>
> As explained in line 148, when sending in $n$ input images, the $i$’s neuron will output $n$ different values. We collect them as an $n$-dimensional activation vector, $\nu_i$. The correlation $ρ_{ij}$ is the correlation between $\nu_i$ and $\nu_j$. As for how the input images are prepared in experiments, please refer to Lines 233 to 238 and Lines 294 to 305.
>
> __Q3.__ Line 168-177: on birth/death time: where the notion of "time" comes from? In which procedure?
>
> __Answer:__ This is standard convention in topological data analysis literature,  as defined in e.g. [17, 18]. Intuitively, as we continuously increase the threshold $t$, more and more simplices are added to the Vietoris-Rips filtration. Intuitively, the complex is “growing” over this process. The changing threshold $t$ is viewed as time.
>
> During this process, topological objects are created at a certain value/time t and disappear at a certain value/time t, which resembles the concept of  birth and death. Then it is natural to talk about the life-time of a feature, namely the time passed between the birth and death (as mentioned in lines 170-171). We will clarify this in the paper.
>
> __Q4.__ "persistence diagrams" or "cycles of high persistence" (Line 184-185) are not defined anywhere in the paper.
>
> __Answer:__ We defined persistence diagrams in line 174. A diagram is a 2D plane   containing dots whose coordinates are birth and death times.  As for “cycles of high persistence”, they are simply loops that correspond to topological features with a relatively high persistence (long lifetime). We will clarify in the paper.
>
> ### Theory
>
> __Q5.__ Theorem 2: This is not much more than a "uniform concentration bound" for a finite function-class.
>
> __Answer:__ Theorem 2 requires not only the concentration bound, but also the stability theorem of persistence diagrams. It provides an important theoretical guarantee to our algorithm; it ensures that the topological difference we observed is real.
>
> __Q6.__ And what is $\psi$? has it ever been defined before?
>
> __Answer:__ Thank you for pointing out the confusion. We will clarify this. $\psi$ is any bi-variate scalar function that has a bounded value. In fact, $\psi$ is meant to be a general form of the covariance function. And $M$ is meant to be a generalized form of the correlation matrix ($\rho$, as defined in Section 3).
>
> ### Experiments
>
> __Q7.__ I don't understand why your experiment should suddenly be in a data-limited setting, while your theory is based on the law of large numbers.
>
> __Answer:__  Our approach follows a scientifically sound strategy in current research on analyzing deep neural networks. Most theoretical guarantees depend on strong assumptions that are unrealistic in practice. So we first empirically verify the theory in a controlled setting strictly following the assumptions. Next, we gradually extend the proposed method to a real-world setting, tracking and verifying the relaxations we make at every step.
>
> Following this practice, we divide our experiments into two steps. In step 1, the analysis in Section 4.2 assumes full access to the data. It was meant to understand the phenomenon in a controlled setting. The results in Fig 2 are fully guaranteed by Thm 2; the observed topological difference between Trojaned and clean models is real.
>
> Building on the success of step 1, in Section 5, we propose a practical solution for the real-world limited-data setting (which is a popular setting in Trojan detection literature). The theoretically-guaranteed analysis results in Section 4.2 give us a “theoretically sound rationale” to apply the topological algorithm in practice, although not fully guaranteed.
>
> Please also note that our theory, even though based on the law of large numbers, is not too demanding in the sample size. The sample complexity only depends on the number of parameters in a logarithmic manner.

---

> > ### Comment · Reviewer_wnSc · 2021-08-28
> > **Post Rebuttal**
> >
> > I still think the concept of persistent homology is vague in a way explained in this paper (and how it is applied to extract features from neural network). Although I initially thought it was because of the lack of rigorous and concise description, now I am more convinced that probably the way it is explained in the paper is good enough as an empirical paper. I increased my score due to this.
> >
> > However, I think my criticism on the writing and theory are still valid - for instance, I think some important measures like $\rho$ should be defined precisely so that any readers can implement it without looking actual code. I believe that this can be a good empirical work if substantially more weights are given to experiments.

---

> > > ### Author Response · Authors · 2021-08-30
> > > **Our Further Response to Reviewer wnSc**
> > >
> > > We really appreciate the reviewer’s response -- we understand his/her position better now regarding the presentation. However, the current criticism is still vague to us. We would like to inquire for further clarification as to how the reviewer assessed our paper, e.g., regarding the originality, quality and significance.
> > >
> > > **"I believe that this can be a good empirical work if substantially more weights are given to experiments.”**
> > >
> > > **A:** Could the reviewer please be more specific about the potential issues with the experiments? If the existing experimental evidence is not sufficient, we would be happy to provide more -- as we did in response to Reviewer 3K3q; the new results show good transfer between network architectures (better than for baseline detectors).
> > >
> > > Perhaps the real misunderstanding is about the scope of this work. Our intention was to show that topology can help distinguish Trojaned networks, as well as the intriguing effect of “Trojan-induced short cuts” observed through topological features. We made no attempt to answer why this approach works from a mathematical standpoint -- which in our view will require a lot of theoretical work (see our response to Reviewer r1eN for possible directions). However, we think the intuitive neurobiological explanation we provided is quite convincing -- and fits NeurIPS well. Having said that, if the reviewer can suggest more concrete experiments to further investigate the phenomenon, we would do our best to incorporate them in the final version.
> > >
> > > **“[...] important measures like ρ should be defined precisely so that any readers can implement it without looking actual code.”**
> > >
> > > **A:** This is an excellent point. Our method works with many types of correlations, and we described it as such (Line 149-150). We will make sure to be very explicit about the type of correlation used in the actual experiments, to facilitate reproducibility. If the reviewer is aware of more issues like this, please do not hesitate to let us know. We are confident we can fix them in the final version.

---

> > > > ### Comment · Reviewer_wnSc · 2021-08-31
> > > > **...**
> > > >
> > > > Thanks to authors for further clarification. I am sorry for the confusion caused by my phrase: by more weights to experiments, I meant it for the presentation - current version gives an impression that the work focuses more on developing theory behind experiments. So yes, it is basically due to the misunderstanding about the scope. I understand that the concept of persistent homology is too complex to be learnt in a single conference paper - but still, I am not sure whether the current explanation is easily reachable to readers who are not quite familiar with the topic (and whether one can implement it purely based on the paper).
> > > >
> > > > Regarding experiments, I re-read the paper, and am now more convinced that conducted experiments are quite convincing after understanding the scope. Now I feel more positive about the work and support the acceptance. Suggesting more concrete experiments is beyond my capability, but I think reviewer 3K3q made some good points.

---

> > > > > ### Author Response · Authors · 2021-09-02
> > > > > **Further Response to wnSc**
> > > > >
> > > > > We wanted to thank the reviewer for the extra effort spent to re-evaluate the paper. We are happy that the misunderstanding was resolved.
> > > > >
> > > > > As for the understandability of the topological part: we will rethink the exposition based on the received comments -- perhaps we can make the paper more widely accessible.
> > > > >
> > > > > Regarding reproducibility, we do agree that it may be hard for a non-expert (in computational topology) to reproduce the method from scratch. Having said that, we will be releasing the entire source code (including the topological part) -- so re-running and tuning the experiments should not be an issue.
> > > > >
> > > > > Overall, we would like to thank the reviewer for the time spent, and his/her flexibility -- and for ultimately supporting our work despite initial (justifiable) objections.

---

> ### Author Response · Authors · 2021-08-27
> **Soliciting Post-Rebuttal Response**
>
> Dear Reviewer wnSc,
>
> Understanding you must be busy, we are wondering whether our initial responses have addressed your concerns regarding presentation. We ask you to kindly let us know if there is any other confusion, which possibly may prevent you from assessing the merits of our method. We would be more than happy to clarify any such doubts during the discussion period. Thank you very much!
>
>
> Sincerely,
>
> Authors

---

### Decision · Program_Chairs · 2021-09-27

**Decision:**

Accept (Poster)

**Comment:**

This is an interesting paper with a creative, deep and promising approach.
Moreover, the thorough rebuttal and subsequent discussion seems to have addressed successfully the majority of the main comments the reviewers have raised.